# MicroRNA-221-3p inhibits the inflammatory response of keratinocytes by regulating the DYRK1A/STAT3 signaling pathway to promote wound healing in diabetes
Keyan Hu[1,2,7], Lei Liu[1,7], Songtao Tang[1,7], Xin Zhang[1], Hongfeng Chang[1], Wenyang Chen[3], Taotao Fan[4], Lesha Zhang[5], Bing Shen ◉[6] ✉ & Qiu Zhang ◉[1] ✉

Diabetic foot ulcer (DFU), a serious complication of diabetes, remains a clinical challenge. MicroRNAs affect inflammation and may have therapeutic value in DFU. Here, we find that an miR-221-3p mimic reduces the inflammatory response and increases skin wound healing rates in a mouse model of diabetes, whereas miR-221-3p knockout produced the opposite result. In human keratinocytes cells, miR-221-3p suppresses the inflammatory response induced by high glucose. The gene encoding *DYRK1A* is a target of miR-221-3p. High glucose increases the expression of DYRK1A, but silencing DYRK1A expression decreases high glucose–induced inflammatory cytokine release via dephosphorylation of STAT3, a substrate of DYRK1A. Application of miR-221-3p mimic to human keratinocytes cells not only decreases DYRK1A expression but also inhibits high glucose–induced production of inflammatory cytokines to promote wound healing. This molecular mechanism whereby miR-221-3p regulates inflammation through the DYRK1A/STAT3 signaling pathway suggests targets and therapeutic approaches for treating DFU.

Diabetes mellitus is one of the most common chronic diseases in the world. In diabetes mellitus, increased fasting blood glucose levels enhance the risk of microvascular and macrovascular diseases, leading to a series of organ dysfunctions and various complications[1,2]. Among these complications, local tissue microcirculation disorders may lead to long-term wound non-healing, a refractory complication of diabetes seriously affecting patient behavior, quality of life, and mortality rate[3]. Diabetic foot ulcers (DFUs) are the most common type of wound-healing disorder in patients with diabetes, with >15% of patients with diabetes expected to develop DFU during their lifetime[4]. Individuals with DFU have a higher rate of amputation, and more than half of persons with amputation die within 5 years[5]. Moreover, the long-term treatment for DFU generates large medical expenses, increasing the financial burden to society[6]. Therefore, discovery of therapeutic targets and strategies for the treatment of DFU is urgent.

Epigenetic inheritance is a common process that regulates gene expression and can thus alter numerous biological processes, including tissue repair[7]. MicroRNAs (miRNAs), approximately 18–25 nucleotides in length, are the most widely studied non-coding RNAs. They play a regulatory role by binding to the 3′untranslated region of specific messenger RNAs to inhibit gene expression[8]. The important role of miRNAs in wound healing has been widely reported[9]. For example, miR-132[10], miR-126,[11] and miR-21[12] have been used therapeutically to promote wound healing in

[1]Department of Endocrinology, The First Affiliated Hospital of Anhui Medical University, Hefei, China. [2]Department of Endocrinology, The First Affiliated Hospital, and College of Clinical Medicine of Henan University of Science and Technology, Luoyang, China. [3]Central Laboratory, Fujian Medical University Union Hospital, Fuzhou, China. [4]Center of Experimental Practice, Anhui Medical University, Hefei, China. [5]School of Basic Medical Sciences, Anhui Medical University, Hefei, People's Republic of China. [6]Dr. Neher's Biophysics Laboratory for Innovative Drug Discovery, State Key Laboratory of Quality Research in Chinese Medicine, Macau University of Science and Technology, Macau, China. [7]These authors contributed equally: Keyan Hu, Lei Liu, Songtao Tang. ✉e-mail: bshen@must.edu.mo; zhangqiu@ahmu.edu.cn

persons with diabetes by regulating the inflammatory response, angiogenesis, collagen reorganization, and re-epithelialization. Previous studies have also shown that miR-221-3p is involved in the regulation of cervical cancer cell migration and invasion[13], lipid metabolism of adipocytes[14], and proliferation of pulmonary artery smooth muscle[15]; however, there are few reports on its involvement in wound healing. A previous study by members of our group found that miR-221-3p significantly promoted wound healing in normal and diabetic mice[16] although the mechanism underlying this effect was not determined.

Dual specificity tyrosine phosphorylation-regulated kinase 1A (DYRK1A) is a gene encoding a kinase with diphosphorylation activity. Although the DYRK1A enzyme is known to act on a wide range of substrates, participates in a variety of cellular processes, including proliferation, apoptosis, metabolism and inflammation, and plays key roles in neurodegenerative diseases, tumors, diabetes, and other diseases[17–19], its involvement in DFU has not been examined. We hypothesized that miR-221-3p regulates the *DYRK1A* gene to ultimately promote wound healing in individuals with diabetes. After first testing whether an miR-221-3p agomir—a chemically modified double-stranded small RNA that mimics endogenous miR-221-3p—would promote wound healing in a mouse model of diabetes, we examined the molecular mechanisms underlying the regulation of the *DYRK1A* gene by miR-221-3p through in vitro and in vivo experiments.

## Results

### Effect of miR-221-3p on wound healing in mice

To determine the effect of miR-221-3p on wound healing in mice, we injected miR-221-3p agomir into the edges of an excisional wound in healthy mice and in mice injected with streptozotocin to induce a mouse model of diabetes (hereafter called diabetic mice). We first confirmed that the diabetic mice had reduced weight ($27.92 \pm 0.56$ vs. $18.7 \pm 0.93$ g, $P < 0.01$), increased glucose levels ($135.36 \pm 6.09$ vs. $433.8 \pm 50.13$ mg/dL, $P < 0.01$), and reduced fasting insulin levels ($11.41 \pm 0.32$ vs.

$3.71 \pm 0.29$ mIU/L, $P < 0.05$) compared with normal mice. In addition, miR-221-3p expression in the epidermis at the wound edge was markedly increased for 1 to 9 days after the injection (Supplementary Fig. 1). Wound healing was significantly delayed in diabetic mice compared with healthy control, and significantly accelerated in both diabetic and healthy control mice after application of miR-221-3p agomir compared with a miRNA mimic negative control (NC) (Fig. 1). In addition, our data indicated that the overexpression of miR-221-3p induced a stronger promotion of wound healing in diabetic mice than that in healthy control mice. These data indicate that treatment with miR-221-3p promotes excisional wound healing in both healthy and diabetic mice.

### Effect of miR-221-3p on the inflammatory response at the edge of skin wounds

Myeloperoxidase (MPO) and CD68 are representative biomarkers of neutrophils and macrophages, respectively. Immunohistochemical staining showed that the expression levels of MPO and CD68 were significantly reduced in the miR-221-3p agomir–treated group compared with the miRNA mimic NC–treated group on days 3, 7, and 11, but not on day 1, after treatment (Fig. 2a–d). We also assessed expression levels of proinflammatory cytokines that are closely related to cell infiltration and migration. Our quantitative polymerase chain reaction (qPCR) results indicated that the mRNA levels of proinflammatory cytokines interleukin (*IL*)-*1β*, *IL-6*, *IL-8*, and tumor necrosis factor alpha (*TNF-α*) were significantly reduced in the miR-221-3p agomir–treated group compared with the miRNA mimic NC–treated group on day 11 after treatment. Among them, *IL-1β* was also significantly decreased on day 1 and day 7, and *IL-8* on day 7 (Fig. 2e–g). We labeled M1-type macrophages with CD86 and M2-type macrophages with CD206. qPCR results indicated that miRNA-221-3p significantly reduced the mRNA expression level of CD86 (Fig. 2h). These results suggest that miR-221-3p may reduce the infiltration of neutrophils and macrophages and the expression of proinflammatory cytokines in

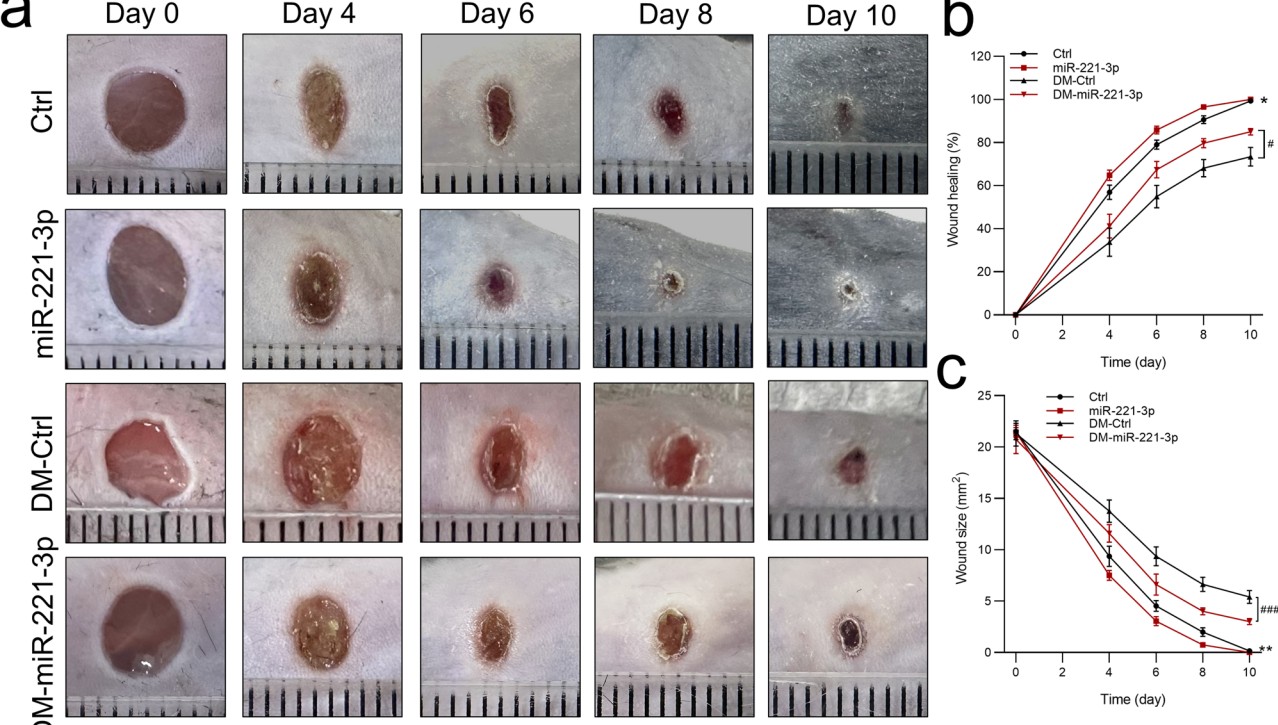

**Fig. 1 | Effects of miR-221-3p on wound healing in normal and diabetic mice.** Representative photomicrographs (**a**) and summary data (**b**, **c**) showing changes in the area of excisional wounds on days 0, 2, 4, 8, and 10 in normal mice injected with either miRNA mimic negative control (Ctrl) or miR-221-3p agomir (miR-221-3p) and diabetic mice injected with either miRNA mimic negative control (DM-Ctrl) or miR-221-3p agomir (DM-miR-221-3p). The smallest scale division of the ruler is 1 mm in a. Data are presented as mean ± SEM, $n = 7$–$8$. *$P < 0.05$, **$P < 0.01$ for Ctrl vs. miR-221-3p. #$P < 0.05$, ##$P < 0.01$ for DM-Ctrl vs. DM-miR-221-3p.

**Fig. 2 | Effects of miR-221-3p on the inflammatory response at the edges of an excisional skin wound in mice.** Representative immunohistochemistry images (**a**, **b**) and summary data (**c**, **b**) showing myeloperoxidase (MPO) (**a**, **c**) and CD68 (**b**, **d**) expression levels on days 1, 3, 7, and 11 after wound formation in diabetic mice injected with miRNA mimic negative control (DM-Ctrl) or miR-221-3p agomir (DM-miR-221-3p). Density of positive cells was calculated using the following formula: number of positive cells/area size ($n$ = 4–6). Scale bars, 50 μm. Summary data showing relative mRNA expression levels of *IL-1β, IL-6, IL-8,* and *TNF-α* at the edge of the skin wound on days 1 (**e**), 7 (**f**) and 11 (**g**) ($n$ = 6–10). **h** Summary data showing relative mRNA expression levels of *CD86* and *CD206* at the edge of the skin wound on days 11 ($n$ = 6–10). The mRNA levels were normalized to 18S ribosomal RNA. Data in c-h are presented as mean ± SEM. Solid circles represent DM-Ctrl group and squares, DM-miR-221-3p group. *$P$ < 0.05, **$P$ < 0.01 for DM-miR-221-3p vs. DM-Ctrl.

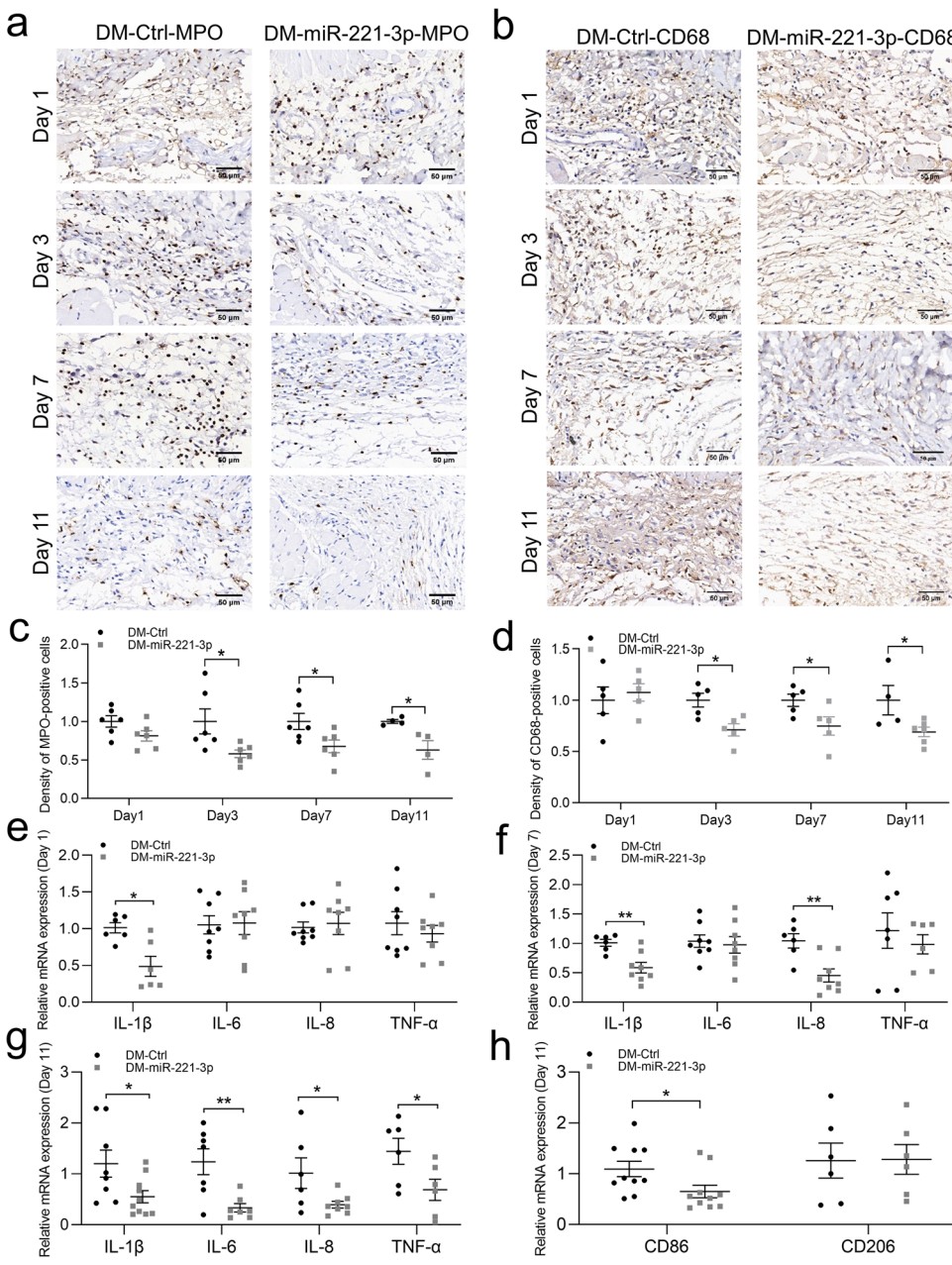

excisional skin wounds of mice, indicating a reduction in the inflammatory response.

## Effect of miR-221-3p on HG-induced inflammatory response of HaCaT cells

To extend our results of the anti-inflammatory effect of miR-221-3p observed in an animal model, we used a human immortalized keratinocyte cell line (HaCaT). We treated these human cells for 24 h with normal glucose (NG, 5.5 mM), high glucose (HG, 35 mM) or an equivalent amount of mannitol as a hyperosmolarity control. The qPCR data showed that the mRNA levels of the proinflammatory cytokines *IL-1β, IL-6, IL-8,* and *TNF-α* were significantly increased in the HG group compared with the NG group (Fig. 3a). In addition, qPCR and enzyme-linked immunosorbent assays (ELISAs) indicated that the mRNA and protein levels of IL-1β, IL-6, IL-8 and TNF-α secreted into the culture medium were significantly increased in the HG group compared with the NG group. This effect was significantly suppressed by miR-221-3p mimic transfection compared with miRNA mimic NC transfection in both the NG and HG group (Fig. 3b–i).

We also performed chemotaxis assays with neutrophils isolated from human peripheral blood. Our results showed that the conditioned supernatant from HaCaT cells overexpressing miR-221-3p suppressed neutrophil migration compared with the supernatant from cells overexpressing NC (Fig. 3j; human neutrophil sorting experiments, Supplementary Fig. 2). Taken together, these results suggest that miR-221-3p suppressed the HG-induced inflammatory response in keratinocytes.

## Role and mechanism of DYRK1A in miR-221-3p suppression of HG-induced inflammatory response in keratinocytes

Because miRNAs act through post-transcriptional regulation of protein-coding genes, identification of their target genes is critical to understanding miRNA functions. Using three major miRNA target gene databases (TargetScan, miRDB, and Encyclopedia of RNA Interactomes [ENCORI]), we identified 239 candidate genes as potential target genes of miR-221-3p (Fig. 4a). To determine the target genes involved in the miR-221-3p effect on HG-induced keratinocyte inflammation, we used RNA sequencing to identify changes in transcript levels. We assessed four groups of data:

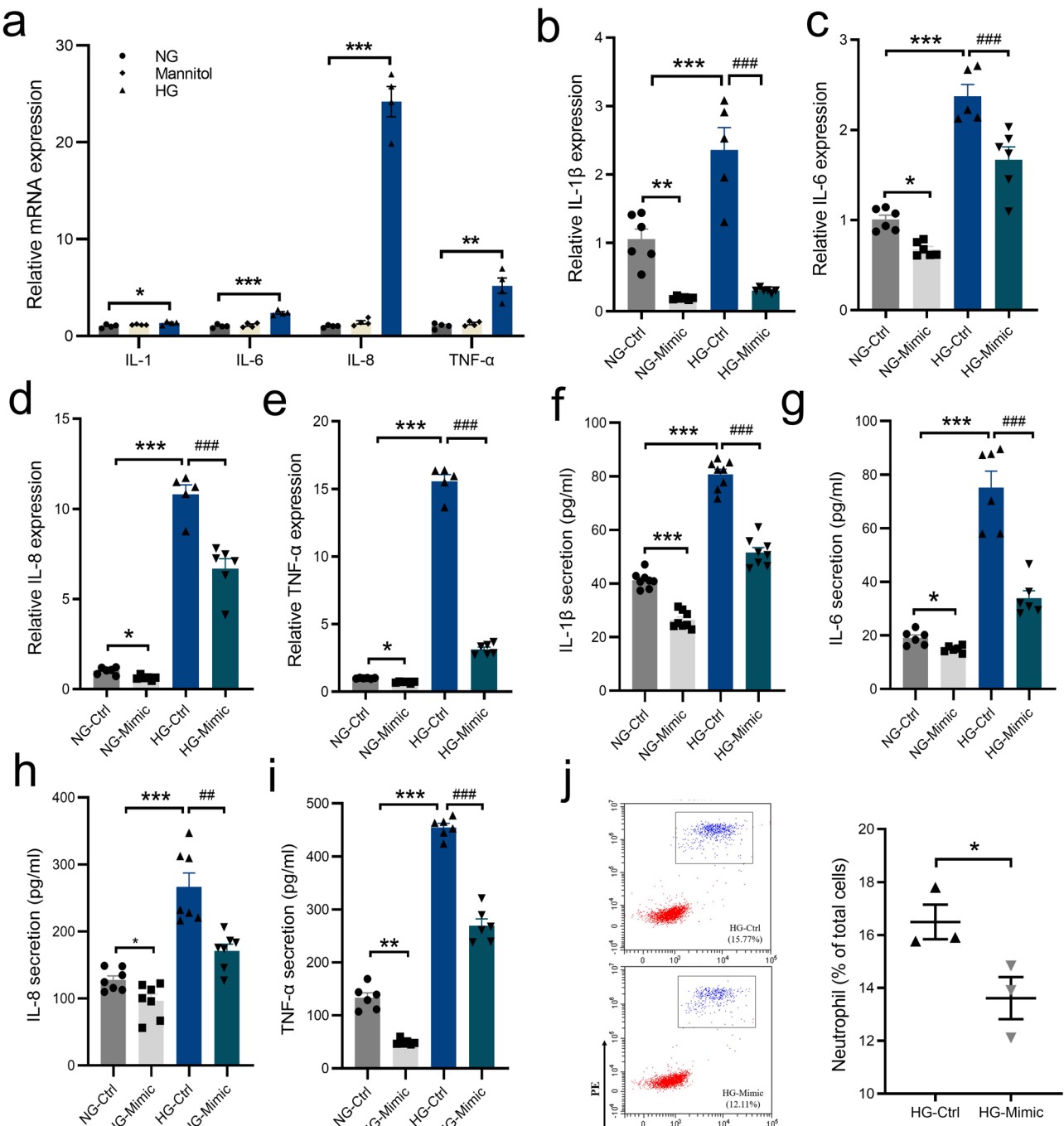

**Fig. 3 | Effects of miR-221-3p on the HG-induced inflammatory response in HaCaT cells. a** Summary data showing relative mRNA expression levels of *IL-1β, IL-6, IL-8,* and *TNF-α* in HaCaT cells treated with normal glucose (NG, 5.5 mM), mannitol (35 mM), or high glucose (HG, 35 mM) (*n* = 4). *$P < 0.05$, **$P < 0.01$, ***$P < 0.001$ for NG vs. HG. Summary data showing relative mRNA expression levels of *IL-1β* (**b**), *IL-6* (**c**), *IL-8* (**d**), and *TNF-α* (**e**) in HaCaT cells cultured in NG or HG medium and transfected with miRNA mimic negative control (Ctrl) or miR-221-3p mimic (Mimic) (*n* = 5–6). *ACTB* was used as the internal reference. Summary data showing protein concentrations of *IL-1β* (**f**), *IL-6* (**g**), *IL-8* (**h**), and *TNF-α* (**i**) secreted into the medium from HaCaT cells cultured in NG or HG medium and transfected with miRNA mimic negative control (Ctrl) or miR-221-3p mimic (Mimic) (*n* = 6–8). **j** Neutrophil chemotaxis toward the conditioned medium from HaCaT cells treated with HG-Ctrl or HG-Mimic was quantified by flow cytometry and is presented as the percentage of total cells that are positive for both CD66b (fluorescein isothiocyanate, FITC) and CD16 (P-phycoerythrin, PE) (*n* = 3). Blue represents neutrophils; red, HaCaT cells. Data (except **j**) are presented as mean ± SEM. Solid circles represent NG-Ctrl group; squares, NG-Mimic group; equilateral triangles, HG-Ctrl group; and inverted triangles, HG-Mimic group. *$P < 0.05$, **$P < 0.01$, ***$P < 0.001$ compared with NG-Ctrl; ##$P < 0.01$, ###$P < 0.001$ for HG-Mimic vs. HG-Ctrl.

downregulated genes in the miR-221-3p mimic–transfected NG group, downregulated genes in the miR-221-3p mimic–transfected HG group, upregulated genes in the HG group compared with the NG group, and the predicted target genes of miR-221-3p. Examining the data using Venn diagrams, we obtained an intersection of these four groups, which contained seven genes: *DYRK1A, THBS1, RALGAPA1, CHD7, KMT2A, KLF7,* and *RREB1* (Fig. 4b). Among them, DYRK1A is involved in the inflammatory response[20]. Thus, we sought to confirm *DYRK1A* as a target gene of miR-221-3p by generating a luciferase reporter construct that contained either the wild-type (WT) 3′ untranslated region (3′-UTR) of the *DYRK1A* gene or

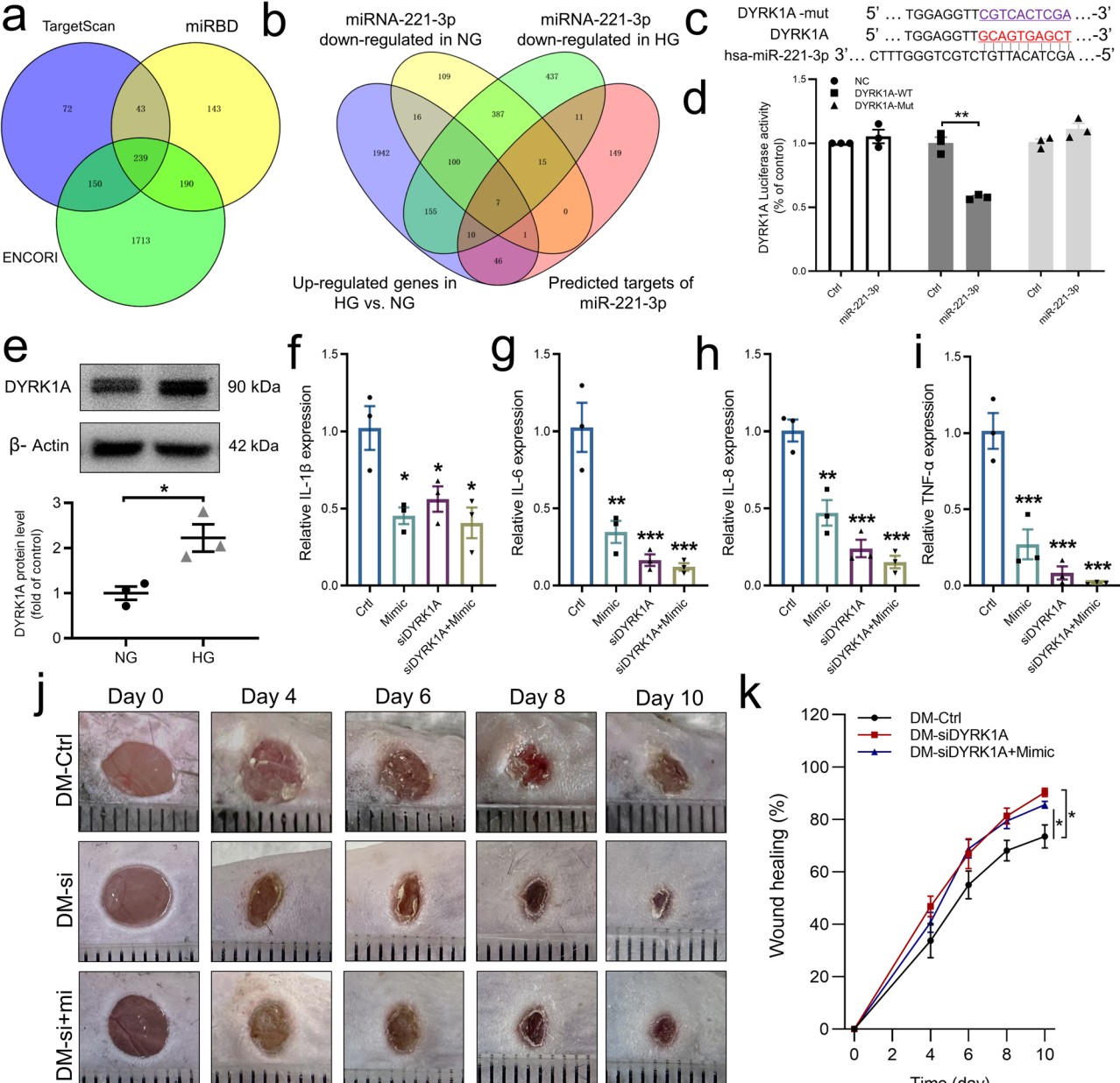

**Fig. 4 | Effect of DYRK1A on HG-induced inflammatory and diabetes wound healing. a** Venn diagram showing the number of potential target genes for miR-221-3p in three major miRNA target gene databases. **b** Venn diagram showing the number of the genes overlapping across the four indicated groups. **c** Sequence alignment of the wild-type (WT) 3′-untranslated region (3′-UTR) of DYRK1A containing putative miR-221-3p-binding sites and the miR-221-3p sequence. Mutated (mut) 3′-UTR of DYRK1A without the putative miR-221-3p-binding site. **d** Relative luciferase activities of plasmids carrying WT or mutant DYRK1A 3′-UTR in HEK293T cells co-transfected with miRNA mimic negative control (NC) or miR-221-3p mimic ($n = 3$) were determined using the Luciferase Dual Reporter gene assay detection system. **e** Representative image and summary data showing protein expression levels of DYRK1A. Protein band intensities were normalized to β-Actin ($n = 3$). Summary data showing relative mRNA levels of *IL-1β* (**f**), *IL-6* (**g**), *IL-8* (**h**) and *TNF-α* (**i**). *ACTB* was used as the internal reference ($n = 3$). Representative photomicrographs (**j**) and summary data (**k**) showing changes in the area of excisional wounds on days 0, 2, 4, 8, and 10 in diabetic mice injected with either negative control (DM-Ctrl), DYRK1A siRNA (DM-si) or both DYRK1A siRNA and miR-221-3p agomir (DM-si+mi). The smallest scale division of the ruler is 1 mm in (**j**). Data are presented as mean ± SEM, $n = 5–7$. *$P < 0.05$ compared with the Ctrl group.

a sequence in which the binding site of miR-221-3p was mutated (Fig. 4c). Compared with miRNA mimic NC transfection, transfection with the miR-221-3p mimic significantly inhibited the luciferase activity of cells co-transfected with the DYRK1A WT 3′-UTR construct, but this inhibitory effect was undetected when the cells were co-transfected with the mutated construct (Fig. 4d).

We next assessed the role of DYRK1A in HG-induced inflammation of HaCaT cells. The expression level of DYRK1A protein was significantly higher in culture medium in the HG group compared with the NG group (Fig. 4e), and transfection with miR-221-3p mimic decreased both DYRK1A mRNA and protein expression levels in HaCaT cells (Supplementary Fig. 3). We then used DYRK1A-specific siRNA to suppress the expression of DYRK1A (Supplementary Fig. 4). The results of our functional study indicated that DYRK1A siRNA strongly reduced mRNA expression levels of the pro-inflammatory cytokines *IL-1β, IL-6, IL-8* and *TNF-α* in miRNA mimic NC- or miR-221-3p mimic–transfected HaCaT cells cultured in HG medium (Fig. 4f–i).

To verify the effect of DYRK1A on wound healing in vivo, we injected DYRK1A siRNA and miR-221-3p agomir into the edges of an excisional wound in diabetic mice. Wound healing was significantly accelerated after application of DYRK1A siRNA only (DM-si) or both application of DYRK1A siRNA and miR-221-3p agomir (DM-si+mi) compared with negative control (DM-Ctrl) (Fig. 4j, k). These data indicate that the inhibition of DYRK1A expression promotes diabetes wound healing.

### Role of miR-221-3p in regulating the STAT3 signaling pathway

Previous studies have reported that signal transducer and activator of transcription 3 (STAT3) is a substrate of DYRK1A[19,21]. Thus, to identify the molecular mechanism underlying the effects of DYRK1A, we used co-immunoprecipitation (Co-IP) assays with HaCaT cell lysates to find that

DYRK1A and STAT3 reciprocally pulled down each other (Fig. 5a). To assess the relationship between DYRK1A and STAT3 in an HG environment, we used DYRK1A-specific siRNA to suppress the expression of DYRK1A. Suppression of DYRK1A expression inhibited the phosphorylation of STAT3 (Tyr 705 and Ser 727) induced by HG (Fig. 5b, d). In addition, DYRK1A siRNA significantly suppressed the expression levels of phosphorylated STAT3 (Tyr 705 and Ser 727) in both miRNA mimic NC– and miR-221-3p mimic–transfected HaCaT cells cultured in HG medium but had no effect on the expression of total STAT3 (Fig. 5c, e). These results suggest that miR-221-3p may suppress its DYRK1A target gene to negatively regulate STAT3 phosphorylation and eventually reduce HG-induced inflammatory response in HaCaT cells.

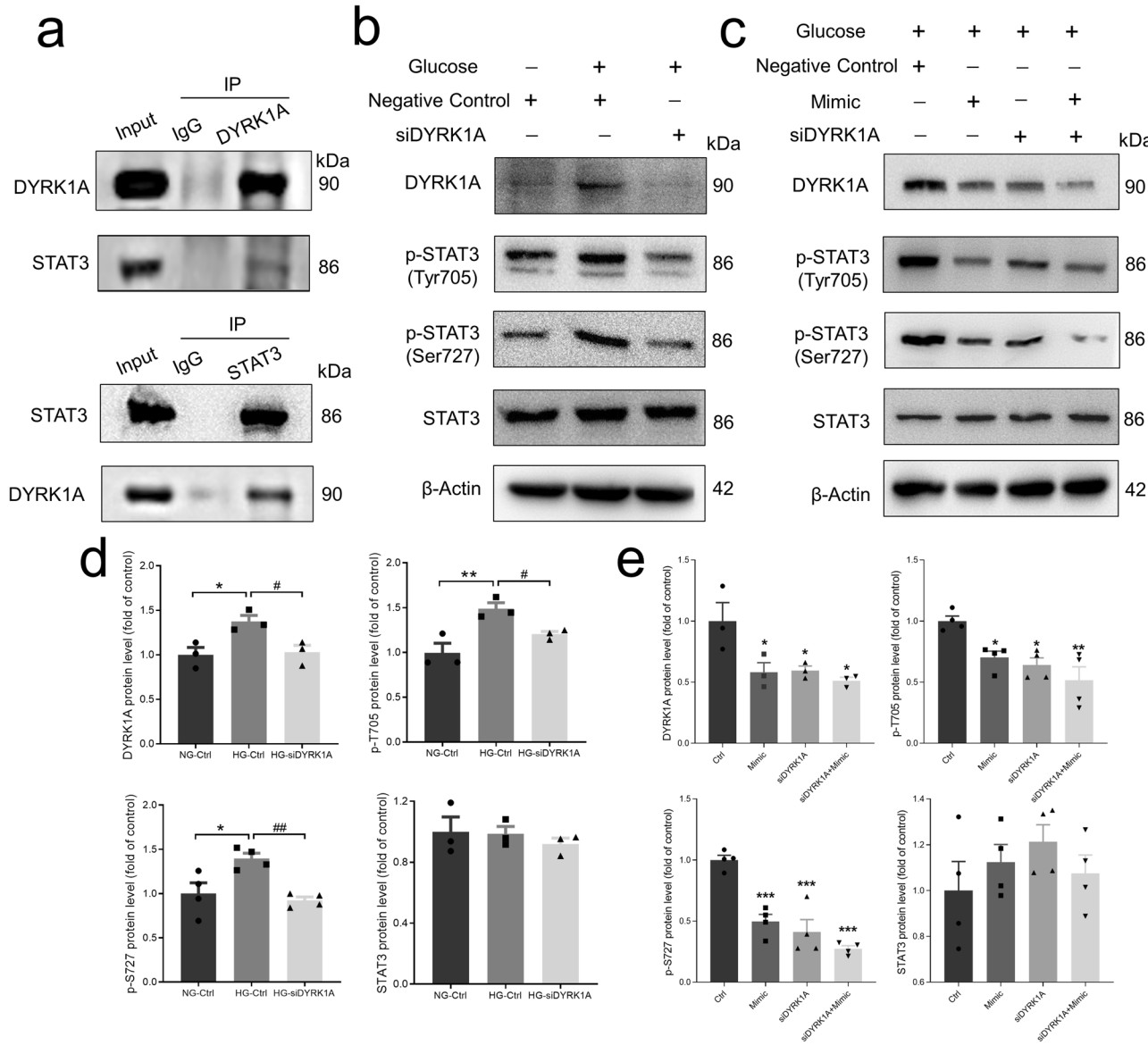

**Fig. 5 | Effect of DYRK1A on the DYRK1A/STAT3 signaling pathway.**
**a** Reciprocal co-immunoprecipitation analysis of DYRK1A and STAT3 in HaCaT cells. Upper panel: cell lysates were immunoprecipitated with an anti-DYRK1A antibody and then detected by an anti-STAT3 antibody; lower panel: cell lysates were immunoprecipitated with an anti-STAT3 antibody and then detected by an anti-DYRK1A antibody. **b, c** Representative images showing protein levels of DYRK1A, phosphorlyated (p)-STAT3 (Tyr705), p-STAT3 (Ser727), and total-STAT3. In (**b**), HaCaT cells were cultured in normal (−) or high glucose (+) medium and transfected with siRNA control (negative control) or DYRK1A siRNA

(siDYRK1A). In (**c**), HaCaT cells were cultured in normal (−) or high glucose (+) medium and transfected with miR-221-3p, siRNA control (negative control), or DYRK1A siRNA (siDYRK1A). **d** Summary data showing the results of (**b**); **e** Summary data showing the results of (**c**); mimic indicates miR-221-3p mimic transfection. Vertically stacked strips of bands in a figure are not derived from the same membrane in any case. All data in (**d**, **e**) are presented as mean ± SEM ($n$ = 3–4). *$P < 0.05$, **$P < 0.01$ compared with NG-Ctrl; #$P < 0.05$, ##$P < 0.01$ for HG-siDYRK1A vs. HG-Ctrl in (**d**). *$P < 0.05$, **$P < 0.01$, ***$P < 0.001$ compared with the Ctrl group in (**e**).

To further explore the underlying molecular mechanisms by which miR-221-3p regulates the inflammatory response, we cultured HaCaT cells with miRNA-221-3p mimic or miRNA mimic NC transfection in NG or HG medium and used RNA sequencing to identify differences in transcript profiles. The results of our Gene set enrichment analysis (GSEA) showed that 98 signaling pathways were upregulated in the HG group compared with the NG group, 33 signaling pathways were downregulated in miR-221-3p mimic–transfected NG group compared with the miRNA mimic NC–transfected NG group, and 31 signaling pathways were downregulated in the miR-221-3p mimic–transfected HG group compared with the miRNA mimic NC–transfected HG group (Supplementary Tables 1–3). To find the key signaling pathways involved in the miR-221-3p effect, we used a Venn diagram of the Kyoto Encyclopedia of Genes and Genomes (KEGG) pathways to find those within the intersection of three groups: upregulated KEGG pathways in the HG group vs. the NG group; downregulated KEGG pathways in the miR-221-3p mimic–transfected HG group vs. the HG group; and downregulated KEGG pathways in the miR-221-3p mimic–transfected NG group vs. the NG group. The intersection contained nine KEGG pathways (Fig. 6a, b).

The JAK-STAT signaling pathway is crucially involved in the inflammatory response[22], and our bioinformatics analysis and experimental data suggested that the JAK-STAT signaling pathway may be involved in the effects of miR-221-3p on HaCaT cells (Fig. 6c). The genes enriched in the JAK-STAT signaling pathway were shown in Supplementary Table 4. Therefore, we used specific inhibitors to examine the role of the JAK-STAT signaling pathway in the miR-221-3p–regulated inflammatory response. The results indicated that signaling inhibitors of JAK (AZD-480, a JAK1/2 inhibitor) abolished miR-221-3p mimic–suppressed expression levels of *IL-6* and *IL-8* in the HG group (Supplementary Fig. 5). In addition, HG in the culture medium of HaCaT cells enhanced the expression levels of phosphorylated STAT3 (Tyr 705 and Ser 727) and the nuclear translocation of STAT3. Application of miR-221-3p mimic significantly suppressed the expression levels of phosphorylated STAT3 (Tyr 705 and Ser 727) in both the NG and HG groups (more strongly in the HG group) and the nuclear translocation of STAT3 (Fig. 6d–g). These results suggest that miR-221-3p suppressed HG-induced inflammation by inhibiting the STAT3 signaling pathway in HaCaT cells.

### Mechanisms underlying the effect of miR-221 knockout on wound healing and the inflammatory response in mice

To determine the effect of miR-221-3p on wound healing in vivo, we first knocked out *Mir221* in mice. Wound closure was significantly delayed in miR-221 knockout (KO) mice compared with WT mice. Similarly, in diabetic mice, skin wound healing was significantly slower in miR-221 KO mice compared with WT mice (Fig. 7a–c).

In diabetic mice, our immunohistochemistry results showed that the expression levels of both MPO and CD68 were increased in the epidermal tissues of the wound on the 11th day after the excisional wounds were created in miR-221 KO mice compared with WT mice (Fig. 8a, b). In addition, DYRK1A protein expression levels were also markedly increased in the epidermal tissues of the wound in miR-221 KO mice compared with WT mice (Fig. 8c, d). Regarding the STAT3 signaling pathway, the expression levels of phosphorylated STAT3 (Tyr 705 and Ser 727) were significantly increased in the wound tissues from miR-221 KO mice compared with WT mice (Fig. 8d). These results suggest that miR-221 KO impaired wound healing and increased the inflammatory response in wounds of diabetic mice.

### Characterization of DYRK1A and STAT3 signaling pathways expression in diabetes mice and human skin wounds

To identify the effects of diabetes on DYRK1A and STAT3 signaling pathways in vivo, the epidermal tissues of the wound on the 10th day after the excisional wounds were collected from normal and diabetic mice. The results showed that the expression level of DYRK1A was increased in the epidermal tissues in diabetes mice compared with normal mice (Fig. 9a–c).

Regarding the STAT3 signaling pathway, the expression levels of phosphorylated STAT3 (Tyr 705 and Ser 727) were significantly increased in the wound tissues from diabetic mice compared with normal mice (Fig. 9a–c).

To study DYRK1A and STAT3 signaling pathways expression in human skin wounds in vivo, we collected the wound edges from acute wound of nondiabetic patient (Ctrl) and DFU. Our results showed that the expression of DYRK1A was upregulated in wounds of DFU compared with the skin from the acute wound (Fig. 9d, e). Regarding the STAT3 signaling pathway, the expression levels of phosphorylated STAT3 (Tyr 705 and Ser 727) were significantly increased in the wound tissues from DFU compared with NC (Fig. 9d, e). These results suggest that DYRK1A and STAT3-signaling pathways play an important role in wound healing in diabetes.

## Discussion

DFU is one of the most common complications of diabetes, and people with diabetes have a 34% lifetime risk of developing DFU[23]. DFU has become a serious global public health problem due to the need for patient hospitalization and even amputation. The primary treatment of DFU includes wound cleaning, revascularization, inflammation control, and decompression[24]. However, clinical treatment is not always effective. New treatments are urgently needed to improve the rate of cure for DFU. Our study showed that (1) an miR-221-3p mimic significantly decreased the time for skin wounds in normal and diabetic mice to heal; (2) an miR-221-3p mimic inhibited inflammatory cell infiltration and immune cell chemotaxis at the edges of skin wounds in diabetic mice; (3) miR-221-3p targeted the gene encoding DYRK1A to regulate STAT3 phosphorylation, thereby inhibiting the inflammatory response caused by an HG environment in HaCaT cells; (4) infiltration of immune cells in the skin at wound edges was increased and wound healing was slowed in diabetic mice with miR-221 knockout; and (5) the expression of DYRK1A and Phosphorylated STAT3 were upregulated in wounds of PU compared with the skin from the DFU. Taken together, these key findings suggest that miR-221-3p may directly inhibit the expression of DYRK1A to regulate the inflammatory response of epithelial tissues mediated by the STAT3 signaling pathway and thus promote skin wound healing in diabetes (Fig. 10). Therefore, miR-221-3p may be a potential therapeutic target warranting development for the treatment of wounds in individuals with diabetes. To our knowledge, this study is the first to identify DYRK1A as a direct target of miR-221-3p and its potential role in wound healing in patients with diabetes.

Wound healing typically occurs in four sequential phases—hemostatic, inflammatory, proliferative, and remodeling—in a continuous and sometimes overlapping manner and involves various cell types, extracellular components, growth factors, and cytokines[25]. Impaired wound healing in individuals with diabetes is the result of multiple factors, including hyperglycemic environment, chronic inflammation, wound infection, vascular insufficiency, hypoxia, and sensory neuropathy. There is an imbalance between pro-inflammatory and anti-inflammatory signals in chronic wounds, with excessive inflammatory mediators leading to delayed resolution of the inflammatory phase, resulting in poor and delayed healing[26,27]. Many cell types involve in wound healing process. For example, neutrophils are recruited in response to cytokines released from damaged and necrotic cells after tissue injury. Endothelial cells participate in the growth of newly formed tissue. Macrophages play an important role in clearing the matrix, cell debris and microorganisms, and fibroblasts participate in the establishment of establishing the extra cellular matrix and collagen deposition related events in wound contraction[28,29]. Keratinocytes are the main cell type in the epidermis layer of the skin. They not only act as an effective barrier but also actively participate in mediating skin inflammatory reactions and interacting with immune cells[30] and play a key role in regulating chronic wound inflammation[31,32]. At present, the most common clinical methods to suppress the inflammatory reaction of DFU include incision decompression, drainage of the abscess, antibiotic treatment, and nanomaterial-based treatment strategies[12,33]. Increasingly more studies are being conducted assessing miRNAs as therapeutic targets or therapeutic methods for wound

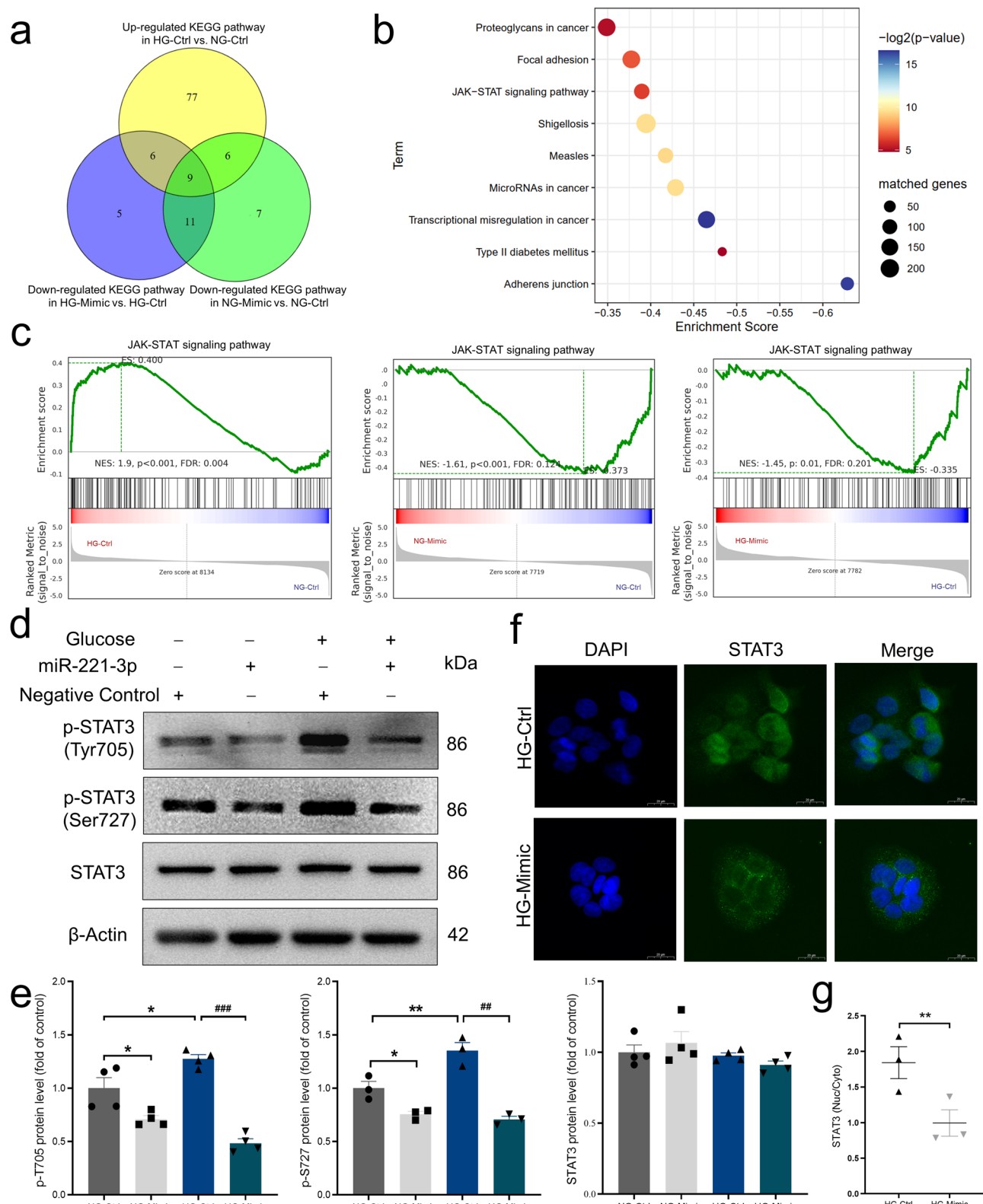

healing in diabetes. For example, Li et al. found that miR-19a/b and miR-20a promote wound healing by regulating the inflammatory response of keratinocytes[34], and Mi et al. showed that miR-146a deficiency delayed wound healing in normal and diabetic mice[35]. In our study, miR-221-3p was shown to be a negative regulator of the keratinocyte inflammatory response. Overexpression of miR-221-3p in HaCaT cells significantly inhibited the expression and release of HG-induced cytokines, including IL-1β, IL-6, IL-8

and TNF-α, and inhibited the transport of neutrophils and macrophages to the wound site. In addition, injection of miR-221-3p agomir into the skin wound edge of diabetic mice significantly inhibited the expression of inflammatory factors and the infiltration of immune cells at the wound edge, and significantly increased the rate of wound healing. By contrast, the rate of skin wound healing in miR-221 KO diabetic mice was significantly slowed, and the skin at the wound edge was accompanied by an obvious

**Fig. 6 | Effects of miR-221-3p on high glucose (HG)-induced JAK-STAT signaling in HaCaT cells. a** Venn diagram showing the numbers of the significantly enriched Kyoto Encyclopedia of Genes and Genomes (KEGG) signaling pathways from Gene Set Enrichment Analysis (GSEA, absolute value of normalized enrichment score ≥1.2, $P < 0.05$) based on RNA sequencing results in the indicated three comparisons. **b** Intersection of enriched KEGG signaling pathways based on three comparisons were indicated. Filled circle size represents the number of enriched genes from each KEGG pathway (larger circles indicate more enriched genes); circle color, $P$ values (colors closer to blue indicate smaller $P$ values). **c** Results of GSEA KEGG analysis showing enriched 'JAK-STAT signaling pathway'. Enrichment plots of 'JAK-STAT signaling pathway' from three comparisons were depicted. A positive normalized enrichment score (NES) indicates gene set enrichment at the top of the ranked list, and a negative NES indicates gene set enrichment at the bottom of the ranked list. FDR, false discovery rate. Green line is the enrichment profile. Dark red in the heat map indicates higher gene expression, whereas blue indicates lower gene expression. The gray area of the heat map represents the signal-to-noise ratio of each gene. **d**, **e** Representative images and summary data showing the protein levels of phosphorylated (p)-STAT3 (Tyr705), p-STAT3 (Ser727), and total-STAT3. In d, HaCaT cells were cultured in normal glucose (NG, −) or HG (+) medium and transfected with miR-221-3p or miRNA mimic negative control (Negative control). In (**e**), Ctrl represents miRNA mimic negative control transfection; Mimic, miR-221-3p mimic transfection. Data are presented as mean ± SEM ($n = 3$–4). *$P < 0.05$, **$P < 0.01$ compared with NG-Ctrl; ##$P < 0.01$, ###$P < 0.001$ for HG-Mimic vs. HG-Ctrl. **f**, **g** Representative immunofluorescence images and summary data showing the nuclear localization of STAT3 in HaCaT cells cultured in high glucose (HG) and transfected with miRNA mimic negative control (Ctrl) or miR-221-3p mimic (Mimic) and the ratio of the intensity of STAT3 in the nucleus (Nuc; shown as blue fluorescence with 4',6-diamidino-2-phenylindole) to that in the cytoplasmic (Cyto; green). Scale bars, 20 μm. Vertically stacked strips of bands in a figure are not derived from the same membrane in any case. Data are presented as mean ± SEM, $n = 3$. Solid circles represent NG-Ctrl group; squares, NG-Mimic group; equilateral triangles, HG-Ctrl group; and inverted triangles, HG-Mimic group. **$P < 0.01$ for HG-Ctrl vs. HG-Mimic.

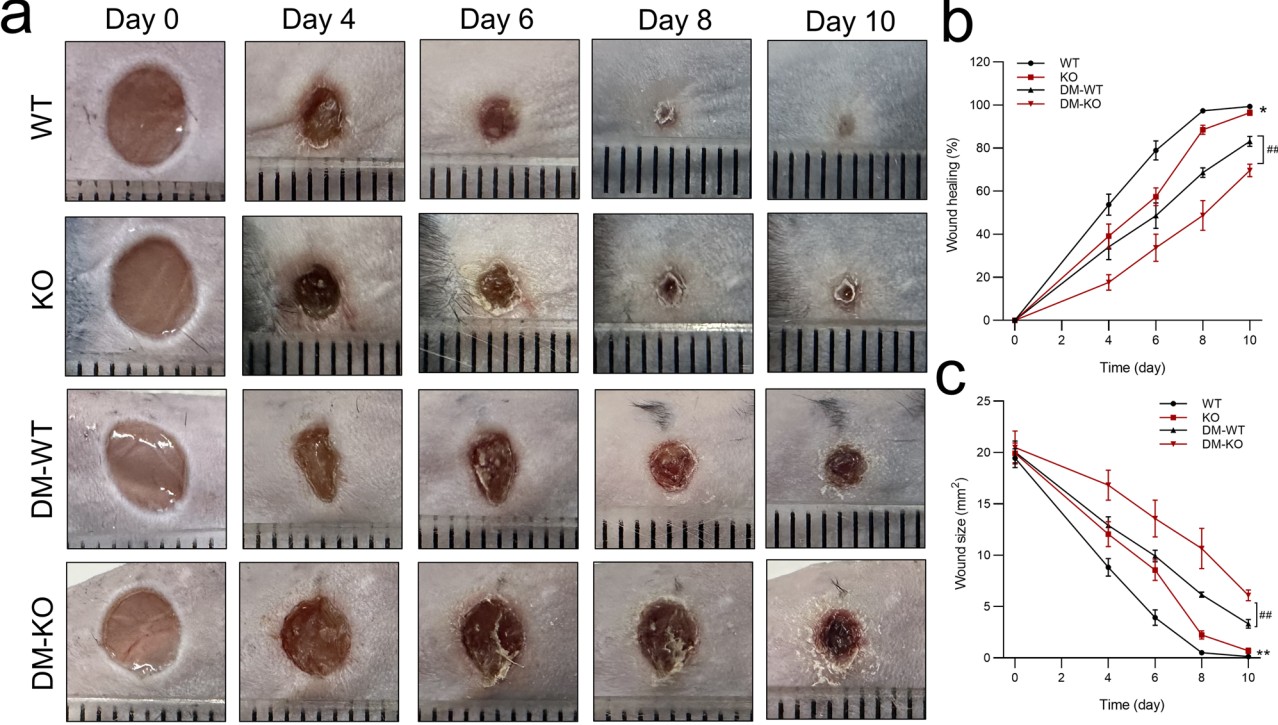

**Fig. 7 | Skin wound healing in diabetic mice with miRNA-221 knockout.** Representative photomicrographs (**a**) and summary data (**b**, **c**) showing wound area changes on days 0, 2, 4, 8, and 10 in wild type (WT) and *Mir-221* knockout (KO) mice and diabetic (DM) WT and KO mice. The smallest scale division of the ruler is 1 mm in (**a**). Data are presented as mean ± SEM, $n = 6$–10. *$P < 0.05$, **$P < 0.01$ for WT vs. KO. ##$P < 0.01$ for DM-WT vs. DM-KO.

inflammatory reaction and leukocyte infiltration. Therefore, our results suggest a potential treatment for skin wounds in patients with diabetes.

miRNAs regulate biological processes by inhibiting mRNA translation or promoting mRNA degradation. Previous studies have reported that miR-221-3p targets *THBS2* to regulate the proliferation and invasion of cervical cancer cells and enhance their metastasis[13]; miR-221-3p induced by inflammation targets *ANGPTL8* to regulate impaired lipid metabolism in metabolic diseases[14]; and miR-221-3p in pulmonary artery smooth muscle cells targets *AXIN2* to induce cell migration and inhibit cell apoptosis[15]. Our study shows that miR-221-3p regulated the STAT3 signaling pathway by targeting DYRK1A, inhibiting the keratinocyte inflammatory response and accelerating wound healing in diabetes. DYRK1A is a protein kinase that plays an important role in a variety of biological processes. Previous studies have shown that DYRK1A inhibitors increase pancreatic β-cell proliferation[36] and are involved in the pathogenesis of Down syndrome[37]. DYRK1A is also involved in regulating the development and progression of

B-cell acute lymphoblastic leukemia[19]. Although DYRK1A may be involved in the development of many diseases, its role in wound healing has not been reported. In this study, we used a high concentration of glucose in the cell culture medium to simulate the high-glucose environment of diabetes, and we used specific siRNA to inhibit DYRK1A expression. Inhibition of DYRK1A expression reduced the synthesis and release of inflammatory factors in keratinocytes caused by a high-glucose environment, which suggested that DYRK1A may be involved in the hyperglycemia-induced inflammatory response in diabetes.

STAT3 is a substrate of DYRK1A[19]. Therefore, we used Co-IP assays to confirm that DYRK1A also interacts with the STAT3 protein in HaCaT cells, which is consistent with previous reports[21]. Our results also showed that HG significantly increased DYRK1A expression and enhanced STAT3 phosphorylation at Try705 and Ser727. Inhibition of DYRK1A expression by specific DYRK1A siRNA reduced STAT3 phosphorylation induced by HG. To further explore the effect of DYRK1A on STAT3 in a

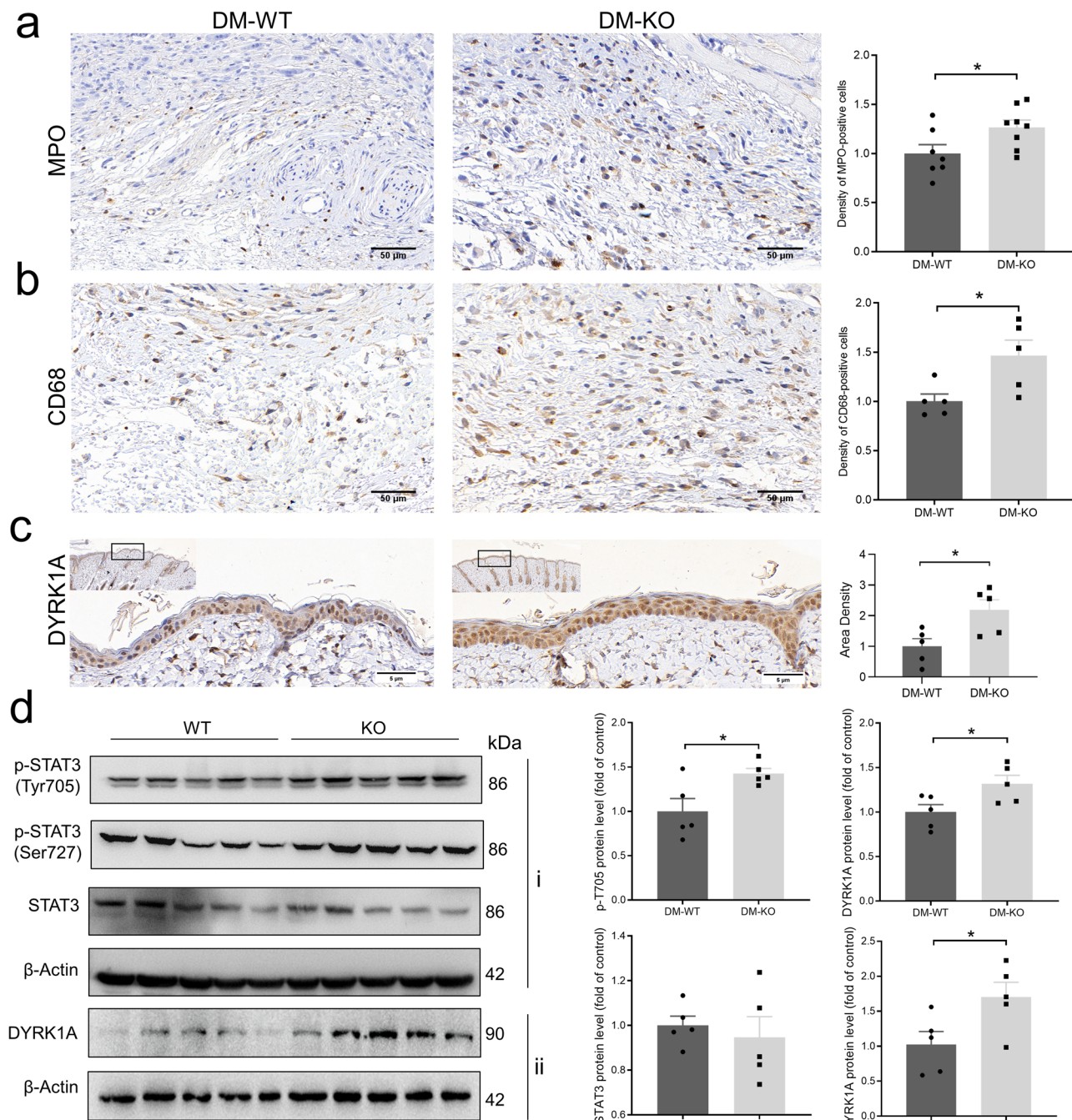

**Fig. 8 | Inflammation and STAT3 signaling pathway changes in skin wounds of *Mir221* knockout mice.** Representative immunohistochemistry images and summary data showing myeloperoxidase (MPO) (**a**), CD68 (**b**) and DYRK1A (**c**) expression levels in skin wound tissues of wild type (WT) and *Mir221* knockout (KO) mice. Data are presented as mean ± SEM (*n* = 5–8). Scale bars, 50 μm in (**a**, **b**), 5 μm in (**c**). **d** Representative images and summary data showing expression levels of phosphorylated (p)-STAT3 (Tyr705), p-STAT3 (Ser727), total STAT3, and DYRK1A in skin wound tissues of WT and *Mir221* KO mice. Vertically stacked strips of bands in a figure are not derived from the same membrane in any case. Data are presented as mean ± SEM (*n* = 5), *P < 0.05.

high-glucose environment, we transfected miR-221-3p mimic into HaCaT cells treated with HG or co-transfected miR-221-3p mimic with DYRK1A siRNA. The results of both experiments showed that miR-221-3p significantly inhibited the phosphorylation of STAT3 at Try705 and Ser727, indicating that miR-221-3p may inhibit the expression of DYRK1A and then inhibit the phosphorylation of STAT3. It is known that the STAT3 signaling pathway is closely related to the pathogenesis of inflammatory and autoimmune diseases. In inflammation-related diseases, cytokines use STAT3 to transduce extracellular signals in cells. Being upstream of the inflammatory signaling pathway, STAT3 is considered to be a

potential therapeutic target for inflammatory diseases[38,39]. Some studies have shown that multiple immune-related signaling pathways, including the IL-6-JAK-STAT3 signaling pathway, are significantly altered in skin tissues from patients with chronic DFU compared with healthy skin tissues[40]. Our results are consistent with those findings. Compared with the acute wound from nondiabetic patients, we found that the expression of DYRK1A and Phosphorylated STAT3 upregulated in DFU, which may contribute to sustained inflammation and impaired healing there. Thus, our study showed that miR-221-3p inhibits the synthesis of inflammatory factors in skin wounds generated in an animal model of diabetes and that the

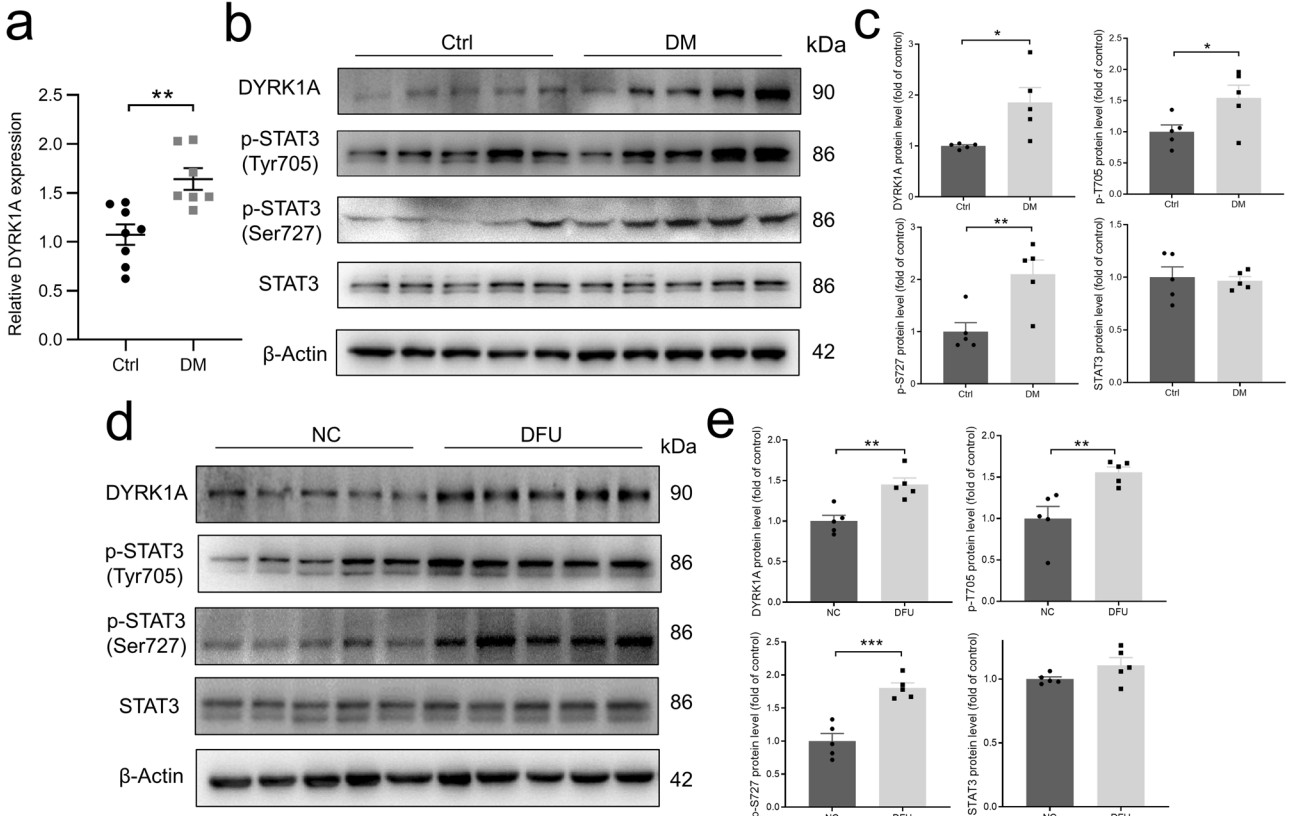

**Fig. 9 | DYRK1A and STAT3 signaling pathway changes in diabetes mice and human skin wounds. a** Summary data showing relative mRNA expression levels of DYRK1A in normal and diabetic mice ($n = 7$–8). **b, c** Representative images and summary data showing protein expression levels of DYRK1A, phosphorylated (p)-STAT3 (Tyr705), p-STAT3 (Ser727), and total STAT3 in normal and diabetic mice ($n = 5$). **d, e** Representative images and summary data showing expression levels of DYRK1A, phosphorylated (p)-STAT3 (Tyr705), p-STAT3 (Ser727), and total STAT3 in skin wound tissues of acute wound and DFU. DFU, Diabetes foot ulcer. Vertically stacked strips of bands in a figure are not derived from the same membrane in any case. Data are presented as mean ± SEM ($n = 5$), *$P < 0.05$, **$P < 0.01$, ***$P < 0.001$.

infiltration of immune cells such as neutrophils and macrophages is mediated through the DYRK1A/STAT3 signaling pathway. The present study provides a molecular mechanism by which miR-221-3p promotes skin wound healing in diabetes. However, due to the limited sample size, the role of DYRK1A and STAT3 signaling pathways in DFU needs to be further verified with more groups and larger sample size.

In conclusion, we demonstrated that miR-221-3p inhibited the inflammatory response and promoted skin wound healing in diabetes by targeting DYRK1A expression and then regulating the DYRK1A/STAT3 signaling pathway. Our study offers a potential target and therapeutic approach for healing skin wounds in patients with diabetes.

## Methods
### Human wound samples
Human samples were collected at the Department of Endocrinology of The First Affiliated Hospital of Anhui Medical University. Patients with DFUs that have not healed and lasted for more than 3 months despite routine treatment were included in this study. The samples in the control group collected from acute wound of nondiabetic patients. Tissue samples were taken using a 6-mm biopsy punch at the nonhealing edges of chronic wounds. All clinical experiments were approved by the Ethics Committee of the First Affiliated Hospital of Anhui Medical University (PJ-2023-04-18). All subjects signed a written informed consent. All ethical regulations relevant to human research participants were followed.

### Animal experiments
All animal experiments were approved by the local authorities and performed in accordance with the guidelines of the Animal Care and Use Committee of Anhui Medical University (LLSC20190426). We have complied with all relevant ethical regulations for animal use. The C57BL/6 WT mice and Mir221 KO mice were generated by and purchased from GemPharmatech (Jiangsu, China). Only male mice were included in this study. Exon 1 of the Mir221-201 transcript (ENSMUST00000083488.3) was used as the knockout region. CRISPR/Cas9 technology was applied to modify the *Mir221* gene. Briefly, the CRISPR/Cas9 system was expressed in fertilized C57BL/6JGpt mouse eggs. Fertilized eggs were transplanted to obtain positive F0 mice, which were confirmed by PCR and sequencing. A stable F1 generation mouse model was then obtained by mating positive F0 generation mice with C57BL/6JGpt mice. Genotypes of KO mice were confirmed by PCR reactions with specific primers (forward: GTCTAACTCTCA-GAAGGATTAGGGTGC; reverse: AGAAGTGGTTAGATTCGTTGG ATCA).

All mice were housed in vented cages with a 12-h light-dark cycle and had access to food and water ad libitum. Eight- to twelve-week-old WT and miR-221-3p KO mice were injected intraperitoneally with streptozotocin (50 mg/kg body weight) (S0130, Boaigang Biological Technology, China) daily for 5 days. Mice in the control group were injected with equivalent doses of citrate buffer solution. Fasting blood glucose levels were checked every 5 days after streptozotocin injection. Mice were considered to have diabetes once fasting blood glucose level was ≥250 mg/dL twice in succession. Twelve weeks after diabetes induction, the mice were used for wound healing study.

To create excisional wounds, the hair on the back of mice was shaved, and a 6 mm biopsy punch was used to produce two full-thickness dermal excisional wounds symmetrically near the dorsal median line. The indicated amount of miR-221-3p agomir (100 nmol/kg) / miRNA mimic NC

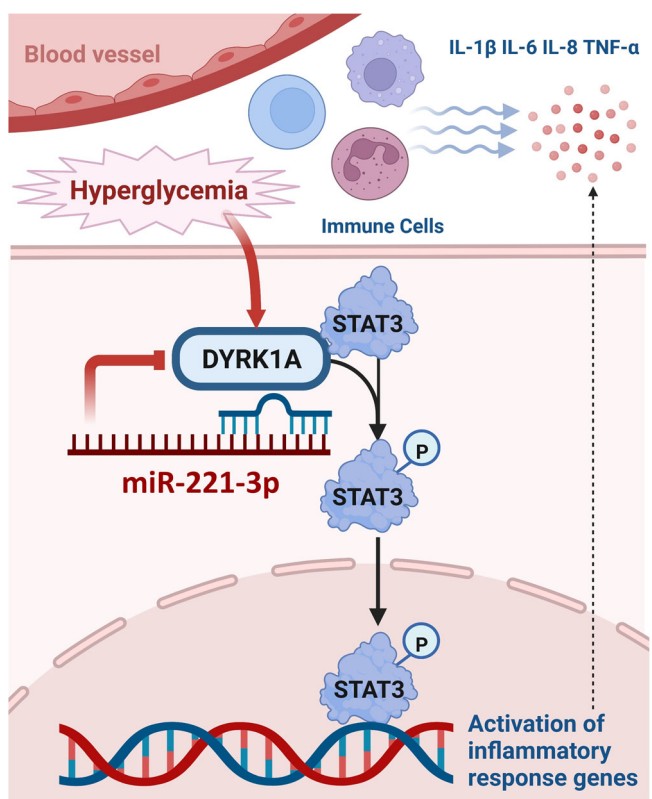

**Fig. 10 | Schematic diagram illustrating the mechanism whereby miR-221-3p influences wound healing.** miR-221-3p inhibited the inflammatory response by targeting DYRK1A expression and then regulating the DYRK1A/STAT3 signaling pathway. Created with BioRender.com.

(GenePharma, China) or DYRK1A siRNA (100 nmol/kg) / siRNA negative control (BioRay, China) was injected intradermally into four points around the wound edges every other day after wounding. The experimental schematic illustration shown in Supplementary Fig. 6. The size of the wound area was photographed and measured every other day before the RNA analogues were injected. Wound area was measured by using a metric ruler that was placed adjacent to the wound. The wound healing rate was calculated as the percentage of the original wound size (assessed using a straight edge ruler) with the following formula: [(initial area−final area)/initial area] ×100%. The tissues were fixed in 10% neutral-buffered formalin for immunohistochemistry assays or snap-frozen in liquid nitrogen and stored at −80 °C for further analysis.

### Immunohistochemical staining
Skin wound samples dissected from mice were fixed in 4% paraformaldehyde overnight at 4 °C, then washed with phosphate-buffered saline (PBS) for 20 min, dehydrated through a graded serious of alcohols (70%, 90%, and 100% ethanol in water), cleared with xylene, and embedded into paraffin. A microtome was used to cut the tissues into 5-μm-thick sections. After deparaffinization and antigen retrieval, sections were blocked by incubation with 0.3% hydrogen peroxide for 30 min. Sections were then incubated overnight at 4 °C with primary antibody CD68 (NB600-985, Novus Biologicals, USA), MPO (AF3667, Abcam, UK) or DYRK1A (DF3270, Affinity Biosciences, China) followed by incubation with a secondary antibody conjugated with streptavidin-horseradish peroxidase. A 3,3′-diaminobenzidine substrate kit was used for color development (PV6001/PV6002, Beijing Zhongshan Golden Bridge Biotechnology, China). Images were captured using a light microscope and analyzed with ImageJ (version 1.8.0) (National Institutes of Health, Bethesda, Maryland, USA). The density of positive cells was calculated using the following formula: number of positive cells/area.

### Cell culture
HaCaT cells were purchased from Pricell (Wuhan, China) and cultured with RPMI 1640 medium (BasalMedia, Shanghai, China) containing 10% FBS and 1% penicillin/streptomycin. After reaching 60% confluence, the cells were stimulated with 5.5 mmol/L glucose or 35 mmol/L glucose for 48 h. Then cells were transfected with 50 nmol/L miRNA mimic NC or miR-221-3p mimic (GenePharma, Shanghai, China), and 50 nmol/L small interfering RNA (siRNA) sequence 1 and 2 of DYRK1A or siRNA NC (GenePharma) for another 48 h using lipofectamine 3000 (L3000015, Thermo Fisher Scientific, USA). The medium was changed every 2 days. HaCaT cell supernatant was collected to measure cytokine content and the cells were harvested for subsequent analysis. The siRNA sequences used in this study are shown in Supplementary Table 5.

### RNA extraction and qPCR analysis
Total RNA from cells and skin wounds were extracted using Trizol (15596-026, Invitrogen, USA), and RNA concentrations were determined with a spectrophotometer (Thermo Fischer Scientific, MA, USA). The RNA was reverse transcribed using cDNA Reverse Transcription Kits (Catalog No. 11141ES60, Yeasen Company, China). Relative gene expression levels were assessed with an ABI QuantStudio6 Pro Real-Time PCR System using a SYBR mixture (11202ES08, Yeasen Company). To quantify the relative level of miR-221-3p, a Hairpin-it™ miRNA RT-PCR Quantitation and U6 normalization Kit (Catalog No. E22005, Genepharma) were used. The $2^{-\Delta\Delta Ct}$ method was applied to calculate the expression levels of mRNA and miRNA relative to the endogenous control genes, β-Actin/18S ribosomal RNA and U6 small nuclear RNA, respectively. Information for all primers/probes used in this study are shown in Supplementary Table 5.

### Enzyme-linked immunosorbent assay
To detect secretion of IL-1β, IL-6, IL-8, and TNF-α in the supernatant, we plated cells in 6-well plates and subjected them to various treatments. After centrifugation at 1000 rpm for 10 min, the supernatant was collected. The levels of secreted IL-1β (E-EL-H0149c), IL-6 (E-EL-H6156), IL-8 (E-EL-H6008), and TNF-α (E-EL-H0109c) (all from Elabscience, China) were determined with ELISA kits according to the manufacturer's instructions.

### Western blotting
Cells or wound tissues were homogenized in radioimmunoprecipitation assay buffer (P0013B, Beyotime Biotechnology, China), and protein concentrations were determined by bicinchoninic acid protein assay (P0010, Beyotime Biotechnology). Equal amounts of protein were loaded, separated by 10% sodium dodecyl sulfate–polyacrylamide gel electrophoresis, and transferred onto polyvinylidene difluoride membranes (Millipore, USA). The membranes were blocked in 5% skim milk at room temperature for 1 h and incubated overnight at 4 °C with primary antibodies. The membranes were then extensively washed in Tris-buffered saline with 0.1% Tween® 20 detergent (TBST) and incubated with secondary antibodies for 2 h at room temperature. After being washed three times with TBST, protein bands were detected using an enhanced chemiluminescence detection system (Bio-Rad Laboratories, Inc.). Luminescence intensity was analyzed using ImageJ. The antibodies used were anti-DYRK1A (YT1435, Immunoway, USA), anti-phospho-STAT3 (Tyr705) (9145S, Cell Signaling Technology, USA), anti-phospho-STAT3 (Ser727) (9134S, Cell Signaling Technology, USA), anti-STAT3 (9139S, Cell Signaling Technology, USA), anti-β-actin (AF7018, Affinity Biosciences, China), and anti-rabbit IgG (A7016, Beyotime Biotechnology, China).

### Co-immunoprecipitation assay
For Co-IP assays, HaCaT cells were lysed with IP buffer (P0013F, Beyotime Biotechnology) followed by centrifugation at 12,000 × *g* for 30 min at 4 °C. The supernatant was aspirated and protein concentrations were determined. Protein A/G Magnetic Beads (HY-K0202, MedChemExpress, USA), together with the corresponding antibodies (dilution 1:100) or normal IgG as a negative control, were incubated at 4 °C for 4 h to form magnetic

bead–antibody complexes. The supernatant was then incubated with the magnetic bead–antibody complexes on a shaker at 4 °C overnight. After that, beads were separated from the cellular proteins by using a magnet and washed in washing buffer. The final bead–antibody complexes were resuspended in 100 µL of loading buffer and subjected to Western blotting as described above.

## Immunofluorescence staining
After transfection, HaCaT cells were fixed with 4% paraformaldehyde for 15 min and then washed in PBS three times for 5 min each time. Permeabilization of the membrane was achieved using 0.3% Triton X-100 in PBS for 30 min followed by washing with PBS three times. Cells were then blocked with 5% bovine serum albumin for 1 h and incubated with primary antibodies overnight at 4 °C. After being washed with PBS three times, the cells were incubated with secondary antibodies (A31572/A21202, Invitrogen) for 1.5 h at room temperature. Cells were then washed with PBS three times for 15 min and stained with 4′,6-diamidino-2-phenylindole, dihydrochloride for 15 min. A confocal laser scanning microscope (LSM 880, Leica, Germany) was used to obtain merged images by superimposing images from different channels.

## Isolation of human neutrophils and neutrophil chemotaxis assay
Human primary neutrophils were isolated from 0.2% EDTA-anticoagulated whole blood collected by venipuncture from a healthy donor. Erythrocytes were removed using Red Blood Cell Lysis Buffer (Tiangen Biotech, China). Isolated cells were stained with anti-human CD66b (305104, Biolegend, China) and anti-human CD16 (302008, Biolegend) for 30 min at 4 °C. Flow cytometry was performed using a Moflo-XDP system (Beckman Coulter), and only double-stained white cells were isolated. Isolated neutrophils were further suspended in RPMI 1640 serum-free medium, and $1 \times 10^6$ cells were added to the inner chamber of a 3-µm polycarbonate membrane cell culture insert (Labselect, China) and incubated with the conditioned medium within 24 h of HaCaT cells being transfected with an miR-221-3p mimic or miRNA mimic negative control. After incubation for 2 h at 37 °C in 5% $CO_2$, the neutrophils that had migrated into the outer chamber were quantified using a CytoFlex Analysis Flow Cytometer (Beckman Coulter).

## RNA sequencing and bioinformatic analysis
Total RNA was extracted as described above. RNA quality was evaluated with a NanoDrop 2000 spectrophotometer (Thermo Fisher Scientific), and the integrity of RNA was analyzed with an Agilent 2100 Bioanalyzer (Agilent Technologies, USA). Libraries were constructed and sequenced using an Ilumina Novaseq 6000 to generate approximately 150 base pair paired-end reads. At least 6 G raw reads for each sample were generated and processed with fastp to acquire clean reads for further analysis using R (version 3.2.0). Mapping of clean reads to the reference genome was conducted with hisat2, and fragments per kilobase of exon model per million mapped fragments of each gene was calculated. Differentially expressed genes were analyzed using DESeq2. A $Q < 0.05$ and a fold change >2 or <0.5 was set as the threshold for indicating a significantly differentially expressed gene. KEGG pathway-based GSEA was performed using GSEA software. Predefined gene sets based on KEGG pathways were used for analysis. All genes detected in RNA sequencing dataset were ranked from the highest to the lowest according to the degree of differential expression between two groups of samples. We then tested whether the genes from the predefined gene sets were enriched at the top or bottom of the ranking list. Enrichment scores and normalized enrichment scores (NES) were calculated using GSEA software. Positive and negative NES indicated higher and lower expression of the KEGG pathway in the treatment group, respectively. Absolute values with NES >1.2, $P < 0.05$, and false discovery rate <0.25 were considered statistically significant.

To search for the predicted targets of miR-221-3p, we employed the miRNA databases TargetScan (http://www.targetscan.org/vert_72/), miRDB (http://www.mirdb.org/), and ENCORI (https://starbase.sysu.edu.cn). We obtained a set of target molecules for miR-221-3p. We then used a Venn diagram to find the intersection of the downregulated target genes in

sequencing results with the predicted target genes to narrow the scope of the target molecules.

## Luciferase reporter assay
Firefly luciferase reporter plasmids containing the 3′-UTR of the DYRK1A gene and an empty luciferase vector were constructed. HEK-293T cells were co-transfected with plasmids containing DYRK1A 3′-UTR WT or mutant fragments and miR-221-3p mimics or miRNA mimic NC using Lipofectamine 3000. Forty-eight hours after co-transfection, luciferase activity was measured using the Dual-Luciferase Reporter Gene Detection kit (E1910, Promega, USA) according to the manufacturer's instructions. Absorbance values were measured with a microplate reader (BioTek Synergy 2, USA). Each experiment was repeated three times.

## Statistics and reproducibility
Data are presented as mean ± SEM. Statistical analysis was conducted using GraphPad Prism software (GraphPad Software, Inc. La Jolla, California). Two-tailed $t$-tests and One-way or two-way analysis of variance were used to determine the statistical significance between groups. For all statistical tests, $P < 0.05$ was considered to be statistically significant. All experiments collected at least three independent samples. Every skin sample from each mouse or subject, or a group of cells from the same well, was defined as one independent sample.

## Reporting summary
Further information on research design is available in the Nature Portfolio Reporting Summary linked to this article.

## Data availability
RNAseq data are available in NCBI SRA database (SRA data: PRJNA956529). The Source data are provided in Supplementary Data 1. Images of uncropped blots are provided in Supplementary Fig. 7. The data underlying this article will be shared on reasonable request to the corresponding author.

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

## Acknowledgements

This work was supported by the National Natural Science Foundation of China (grants No. 82370836, No. 81970703 and No. 82270884). National Natural Science Foundation of China Regional Innovation and Development Joint Fund (grant No. U22A20272). The authors acknowledge the use of BioRender.com to create the schematic diagram in Fig. 10 and Supplementary Fig. 6.

## Author contributions

K.Y.H. contributed to the experimental design, performance, data analysis, and manuscript preparation. L.L. and S.T.T. contributed to the experimental design, data analysis, and manuscript revision. X.Z., H.F.C., W.Y.C., and T.T.F. were involved in conducting the experiments and specimen collection. L.S.Z. contributed to data analysis and discussion of the results. B.S. and Q.Z. supervised the studies and contributed to manuscript preparation.

## Competing interests

The authors declare no competing interests.
