## [Peer Review File · Communications Biology]

Reviewers' comments:

Reviewers #1 & 2 (Remarks to the Author):

In this report Hu et al investigated the effects of MicroRNA-221-3p on wound healing both on normal and diabetic mice. They investigated the molecular mechanisms of MicroRNA-221-3p action using human immortalized keratinocytes and uncovered previously unappreciated mechanisms that involves DYRK1A and STAT3. While the article is well written and protective effects of MicroRNA-221-3p in wound healing are well documented (in this and previous manuscript of the authors) the molecular mechanisms of this effect are not well characterized in in-vivo model, and there are a number of issues in the manuscript that the authors should address.

1) For the comparative analysis of wound healing, representing the normal and diabetic wound healing on the same graph will be beneficial for the manuscript. While diabetic mice heal slower than otherwise healthy mice, nevertheless the direct comparison will be more informative.

2) While the authors mentioned two consecutive 250+ glucose reading, the authors should show the actual glucose levels of these mice in comparison with control mice. Better yet, the insulin levels should be shown too.

3) The authors should characterize the levels of DYRK1A protein and mRNA in mice as well (normal vs diabetic wound, and perhaps normal keratinocytes vs healing), and the entire signaling cascade as they did in HaCaT cells. For example, in their Figure 7D, it seems that 3 out of 5 animals in WT group do not have detectable DYRK1A, which is not reflected in the quantification. Furthermore, the quantification does not make sense, it says fold over control, what is the control? If it's the WT, it should be normalized to 1, but none of the quantifications align to 1.

4) The title of the manuscript is a little misleading, as it seems that MicroRNA-221-3p might have a bigger effect on wound healing in healthy mice. In my opinion more fitting title will drop the diabetes part out.

5) The labeling of the figures should be more detailed, for example, Fig 2C and D need to have the label of antibody, otherwise they all look the same.

6) The authors should justify using 35mmol glucose in HaCaT cells, as this is extremely high level of glucose even for uncontrolled diabetes.

7) Figure 4E. The authors should include the data points in the graph and not only the summary.

8) The authors should explain the difference between pressure ulcer and DFU.

9) Figure 2A/B are too blurry and zoomed out to observe the authors claims made for these data. The authors should include higher quality photos that are more zoomed in to better make their point. Furthermore, the claim that there are fewer migrated immune cells in the treated group based on these H&E data does not seem substantiated. Unless additional photos are included that better support this claim, this assertion should be removed from the text.

10) The authors should present what genes are being enriched in the JAK STAT signaling pathway for their KEGG analysis in 5B.

11) The figure legend for Figure 5C states that this is "a Gene Set Enrichment Analysis of the genes related to the JAK-STAT signaling". If this description is accurate, this is an incorrect use of this analysis and should be removed. GSEA should be applied to the entire gene set, not an already JAK-STAT enriched set of genes. Overall, the methods of this analysis are unclear, and the authors need to be more forthcoming with their approach and ensure it is being conducted in the correct way.

12) Minor point: In the text, authors claim that MPO and CD68 are increased in the epidermal tissue as supported by data from figure 7A/B. However, 7A/B are dermal tissue. The authors need to either present epidermal data or correct the language.

13) The authors claim in the text that DYRK1A was upregulated in wounds PU compared to DFU. However, the data from figure 8A/B seems to directly contradict this. The text and data similarly oppose one another regarding figure 8C/D. In addition to these issues, the relation of DFU biology to PU biology is unclear. A better comparison for DYRK1A staining would be histologically normal skin

from a non-diabetic patient as well as a diabetic patient and compare this to a DFU to show the specificity of the effect of DYRK1A to this physiological context.

Reviewer #3 (Remarks to the Author):

Comments to authors

This original research article was about the mechanism of miR-221-3p regulating inflammation via DYRK1A/STAT3 signaling pathway on diabetic foot ulcer treatment. And they are novel in this diabetic complication therapy. After carefully reading, there was a clear mind for understanding the aims of writing and experimental process. However, there are some major issues, which need to be discussed.

Issues

1. In Fig.2g, the relative mRNA expression levels of IL-1 β , IL-6, IL-8, and TNF- α from different animals were different. And there were significant deviation in the standard deviation of several measurements. It's better to give the explanation of this results.
2. In Fig.7d, there were no bands of DYRK1A in skin wound tissues of first three mice in WT, but the other two with the significant bands. Could you please explain this difference?

Reviewers #4, 5, & 6 (Remarks to the Author):

This is an interesting manuscript investigated the molecular mechanisms underlying the regulation of the DYRK1A gene by miR-221-3p both in vitro and in vivo.

1. At what age of diabetic mice were the wounds created? Fig. 1 shows no difference in wound healing between DM and non-DM mice; this probably means DM mice did not have DM long enough to develop impaired wound healing. Longer duration of diabetes would be needed to study impaired wound healing.
2. The reached conclusions about diabetes are not sustainable since results from the current study shows similar effect in wound healing in DM and non-DM mice. The only reasonable conclusion is that microRNA-221-3p affects wound healing regardless the presence of DM or not.
3. It would be beneficial to use also biomarkers for M1 and M2 macrophages, in addition to neutrophils and macrophages representative biomarkers that were used.
4. It would be helpful to determine the protein levels of pro-inflammatory cytokines in addition to the mRNA expression.
5. More time points are needed for more comprehensive study of the expression levels of pro-inflammatory cytokines (not only day 11).
6. A concentration 35mM Glucose used in the study is too high. A lower glucose concentration could provide more physiologically appropriate.

7. For how many days did the miR-221-3p agomir and the miRNA mimic NC were injected to the wounds? It would be helpful if this information was added to the materials and methods section.
8. It would be beneficial to study/measure the re-epithelization and wound construction from MTS or H&E images, in addition the wound size measurements in fig. 1.
9. In lines 222-227 is mentioned that is described the Fig.5e—j. However, fig.5 does not have h, i and j, but stops at g. Please correct in the manuscript that in line 227 is described the fig. 5d-g.
10. Author used DYRK1A-specific siRNA to suppress the expression of DYRK1A in vitro to further validate the impact of DYRK1A protein on inflammation and also showed the synergistic effect on inflammation reduction when apply both siRNA and miRNA. It looks like applying both siRNA and miRNA has a great potential, have author tried this combination or siRNA (solely) in vivo study?
11. Other studies have shown DYRK1A present in many different cell types, such as macrophage, fibroblasts, and endothelial cells, and have very different impact on angiogenic response. It would be helpful to add more explanation on why keratinocytes was selected in this study and what is the correlation between keratinocytes associated DYRK1A and DFU.
12. Endothelial cells and fibroblasts are important for wound healing and affected by diabetes. It would be helpful to add more discussion on the effect on angiogenesis and collagen formation.

Point-to-point Response

General response: We kindly thank reviewer for your constructive suggestions. In response to your comments, we have followed your suggestions to address these concerns. We believe that this version of the manuscript is largely improved.

Comment 1: For the comparative analysis of wound healing, representing the normal and diabetic wound healing on the same graph will be beneficial for the manuscript. While diabetic mice heal slower than otherwise healthy mice, nevertheless the direct comparison will be more informative.

Response: Thank you very much for your constructive suggestion. In order to highlight the obvious promotion of miRNA-221-3p on wound healing in diabetic mice, we extended the diabetes duration (from the original 4 weeks to 12 weeks) in the new supplementary experiment and represented the normal and diabetic wound healing on the same graph (**Fig.1a** for representative photomicrographs and **Fig.1b, c** for summary data). We hope that the results of the new experiments can provide more information to demonstrate the promotion of miRNA-221-3p on wound healing in vivo.

Comment 2: While the authors mentioned two consecutive 250+ glucose reading, the authors should show the actual glucose levels of these mice in comparison with control mice. Better yet, the insulin levels should be shown too.

Response: Thank you for your suggestion to make experimental results more complete. We added the results of actual glucose levels and the insulin levels of these mice in comparison with control mice to the revised manuscript (on lines 98-101, tracked changes). These results can more strongly prove the successful establishment of diabetic mouse model.

Comment 3: The authors should characterize the levels of DYRK1A protein and mRNA in mice as well (normal vs diabetic wound, and perhaps normal keratinocytes vs healing), and the entire signaling cascade as they did in HaCaT cells. For example, in their Figure 7D, it seems that 3 out of 5 animals in WT group do not have detectable DYRK1A, which is not reflected in the quantification. Furthermore, the quantification does not make sense, it says fold over control, what is the control? If it's the WT, it should be normalized to 1, but none of the quantifications align to 1.

Response: (1) We are sorry for the error in the data quantification and we have re-calculated all the data to the correct method. (2) In the revised manuscript, we characterized the levels of DYRK1A mRNA and protein in normal and diabetic wound in mice (**Fig. 9a-c**). Regarding the STAT3 signaling pathway, the expression levels of phosphorylated STAT3 (Tyr 705 and Ser 727) and Total STAT3 were characterized (**Fig. 9b, c**) (text on lines 272-279, tracked changes).

We hope that these in vivo experiments will make the research more complete. (3) We attribute the absence of DYRK1A to poor antibody effectiveness to mouse species. We purchased a new antibody (YT1435, immunoway, USA) and re-prepared the sample to repeat the experiment in Figure 7d. The repeated experimental results are shown in **Figure 8d** in the revised manuscript.

Comment 4: The title of the manuscript is a little misleading, as it seems that MicroRNA-221-3p might have a bigger effect on wound healing in healthy mice. In my opinion more fitting title will drop the diabetes part out.

Response: First of all, we agree with your point that miRNA-221-3p can promote wound healing in both normal and diabetic mice. In previous results, diabetic mice did not have diabetes long enough to develop impaired wound healing compared with normal mice, which resulted in no significant difference between the promotion effect of miRNA-221-3p on wound healing in normal mice and in diabetic mice. To solve this problem, we extended the duration of diabetes in mice and represented the normal and diabetic wound healing on the same graph for the comparative analysis. In order to reflect the wound healing more intuitively, we also show the change in the size of the wound, the results showed that miRNA-221-3p have a larger effect on wound healing in diabetic mice (**Fig.1c**).

Comment 5: The labeling of the figures should be more detailed, for example, Fig 2C and D need to have the label of antibody, otherwise they all look the same.

Response: Thank you for your suggestion, we have added the label of antibody in the revised manuscript and figures (**Fig. 2a-d**) to make the labeling of the figures more detailed.

Comment 6: The authors should justify using 35mmol glucose in HaCaT cells, as this is extremely high level of glucose even for uncontrolled diabetes.

Response: We agree with your point that 35mM is too high level of glucose. Due to the complex pathological environment in vivo, maybe different patient has different glucose level and different localized tissue has different glucose level. Therefore, many studies have used different glucose to mimic different situation. For example, Rebecca Ward et al. used 50mM glucose to explore the relationship between hippocampal cells in diabetes and inflammasome^[1]. There are also relevant studies on keratinocytes and diabetic wound healing, Son DH et al. used 50mM glucose, found a novel peptide promotes antioxidant gene expression and wound healing in HaCaT Cells^[2]; Cheng TL et al. used 25mM and 50mM glucose found that high-glucose environment reduces the expression of Thrombomodulin in keratinocytes and released soluble Thrombomodulin can promote wound healing under diabetic condition^[3]. We tried to experiment with 25mM, but this glucose level was not enough to damage keratinocytes and cause increased expression of inflammatory factors.

Finally, we chose 35mM glucose for further study and added equivalent amount of mannitol as a hyperosmolarity control to avoid the influence of high osmotic pressure on the experiment (**Fig. 3a**). We have added references to the revised manuscript to justify the glucose level.

Comment 7: Figure 4E. The authors should include the data points in the graph and not only the summary.

Response: Thank you for your suggestion, we have changed the summary into the data points in the graph (**Fig. 4e**).

Comment 8: The authors should explain the difference between pressure ulcer and DFU.

Comment 13: The authors claim in the text that DYRK1A was upregulated in wounds PU compared to DFU. However, the data from figure 8A/B seems to directly contradict this. The text and data similarly oppose one another regarding figure 8C/D. In addition to these issues, the relation of DFU biology to PU biology is unclear. A better comparison for DYRK1A staining would be histologically normal skin from a non-diabetic patient as well as a diabetic patient and compare this to a DFU to show the specificity of the effect of DYRK1A to this physiological context.

Response: Please allow us to combine these two questions to reply. We apologize for the error that the text description did not match the image results, this error has been corrected in the new manuscript.

We agree with your point of view, pressure ulcer and DFU is not the best option to determine the role of DYRK1A in diabetes. Your suggestion is very meaningful, and we have tried our best to repeat the experiment according to the groups you suggested. However, we were unable to collect enough histologically normal skin from healthy person (not approved by the hospital ethics committee). In order to solve this problem as much as possible, we have referred to previous research and finally decided to choose acute wounds from nondiabetic patients^[4-6]. The experimental results of re-collecting samples are shown in **Fig. 9d, e** (text on lines 281-289, tracked changes). We have added limitations due to insufficient sample size to the discussion and will continue to complete the experiment in the future (text on lines 399-401, tracked changes).

Comment 9: Figure 2A/B are too blurry and zoomed out to observe the authors claims made for these data. The authors should include higher quality photos that are more zoomed in to better make their point. Furthermore, the claim that there are fewer migrated immune cells in the treated group based on these H&E data does not seem substantiated. Unless additional photos are included that better support this claim, this assertion should be removed from the text.

Response: According to your suggestion, we recollected samples and repeated H&E staining. Unfortunately, we did not obtain enough slices of the

complete wound for statistical analysis, so the results from H&E staining were removed from the text. In subsequent experiments, we used markers of neutrophils and macrophages for immunohistochemistry experiments to verify the effect of miRNA-221-3p on immune cell.

Comment 10: The authors should present what genes are being enriched in the JAK STAT signaling pathway for their KEGG analysis in 5B.

Response: As you kindly suggested, we added all the genes enriched in the JAK-STAT signaling pathway in **supplemental materials Table 3**.

Comment 11: The figure legend for Figure 5C states that this is “a Gene Set Enrichment Analysis of the genes related to the JAK-STAT signaling”. If this description is accurate, this is an incorrect use of this analysis and should be removed. GSEA should be applied to the entire gene set, not an already JAK-STAT enriched set of genes. Overall, the methods of this analysis are unclear, and the authors need to be more forthcoming with their approach and ensure it is being conducted in the correct way.

Response: We are sorry that our description is incomplete and we would like to explain the process in detail. First, in this study, we performed Gene Set Enrichment Analysis (GSEA) based on three comparisons: HG-Ctrl vs NG-Ctrl, NG-Mimic vs NG-Ctrl, HG-Mimic vs HG-Ctrl. For GSEA analysis, we applied two categories of genes: all genes detected in each sample from RNA-seq, and genes from predefined sets (genes from each KEGG pathway in this study). Fold changes of all gene expression levels between two groups were calculated and sorted from the highest to the lowest. For each gene set (genes from each KEGG pathway in this study), an enrichment score (ES) and normalized enrichment score (NES) were calculated to determine whether genes included in a specific gene set was enriched at the top or bottom of the ranked list of all genes present in the RNA-seq dataset. Gene sets with absolute value of normalized enrichment score ≥ 1.2 and $P < 0.05$ were chosen and considered significantly enriched. Enriched KEGG pathways were summarized in Supplementary Table 2 a-c. Among these enriched KEGG pathways from GSEA, enrichment plots of ‘JAK-STAT signaling pathway’ were shown in Figure 5C. From the above description, it can be found that we have actually applied all genes detected in each sample and genes from predefined gene sets (genes from every KEGG pathway in this study) to GSEA. In other words, for the enrichment plots of ‘JAK-STAT signaling pathway’, overall human genes detected in our RNA-seq dataset and all genes included in “JAK-STAT signaling pathway” detected in our RNA-seq dataset, instead of an already JAK-STAT enriched set of genes, were used to calculate ES and NES values. We have revised the corresponding content in the manuscript. We hope that our revision will make the description more accurate and less misleading (text on lines 827-833, tracked changes).

Comment 12: Minor point: In the text, authors claim that MPO and CD68 are increased in the epidermal tissue as supported by data from figure 7A/B. However, 7A/B are dermal tissue. The authors need to either present epidermal data or correct the language.

Response: Thank you for pointing out this error, **Fig8. a, b** is dermal tissue. We misdescribed it, and we have made corrections in the manuscript.

References

- [1] Ward R, Ergul A. Relationship of endothelin-1 and NLRP3 inflammasome activation in HT22 hippocampal cells in diabetes. *Life Sci.* 2016. 159: 97-103.
- [2] Son DH, Yang DJ, Sun JS, et al. A Novel Peptide, NicotinyI-Isoleucine-Valine-Histidine (NA-IVH), Promotes Antioxidant Gene Expression and Wound Healing in HaCaT Cells. *Mar Drugs.* 2018. 16(8).
- [3] Cheng TL, Lai CH, Chen PK, et al. Thrombomodulin promotes diabetic wound healing by regulating toll-like receptor 4 expression. *J Invest Dermatol.* 2015. 135(6): 1668-1675.
- [4] Ramirez HA, Pastar I, Jozic I, et al. Staphylococcus aureus Triggers Induction of miR-15B-5P to Diminish DNA Repair and Deregulate Inflammatory Response in Diabetic Foot Ulcers. *J Invest Dermatol.* 2018. 138(5): 1187-1196.
- [5] Schultz GS, Sibbald RG, Falanga V, et al. Wound bed preparation: a systematic approach to wound management. *Wound Repair Regen.* 2003. 11 Suppl 1: S1-28.
- [6] Morton LM, Phillips TJ. Wound healing and treating wounds: Differential diagnosis and evaluation of chronic wounds. *J Am Acad Dermatol.* 2016. 74(4): 589-605; quiz 605-6.

Point-to-point Response

General response: We kindly thank reviewer for your constructive suggestions. In response to your comments, we have followed your suggestions to revise the manuscript carefully. We believe that this version of the manuscript is largely improved.

Comment 1: In Fig.2g, the relative mRNA expression levels of IL-1 β , IL-6, IL-8, and TNF- α from different animals were different. And there were significant deviation in the standard deviation of several measurements. It's better to give the explanation of this results.

Response: Thank you for your comment and question. We re-prepared the sample then repeated the experiment, and found that although the standard deviation was reduced, there were still a little deviation. We attribute it to the large individual differences among animals. We chose a large sample size to avoid the influence of individual differences on the experimental results. The new experimental results are shown in **Figure 2g**.

Comment 2: In Fig.7d, there were no bands of DYRK1A in skin wound tissues of first three mice in WT, but the other two with the significant bands. Could you please explain this difference?

Response: Thank you very much for your constructive suggestion. We attribute the absence of DYRK1A to low sensitivity of antibody against mouse samples. We purchased a new antibody (YT1435, immunoway, USA) and re-prepared the sample to repeat the experiment in Figure 7d. The repeated experimental results are shown in **Figure 8d** in the revised manuscript. We hope that will make the research more complete.

Point-to-point Response

General response: We kindly thank reviewer for your constructive suggestions. In response to your comments, we have followed your suggestions to revise the manuscript carefully. We believe that this version of the manuscript is largely improved.

Comment 1: At what age of diabetic mice were the wounds created? Fig. 1 shows no difference in wound healing between DM and non-DM mice; this probably means DM mice did not have DM long enough to develop impaired wound healing. Longer duration of diabetes would be needed to study impaired wound healing.

Comment 2: The reached conclusions about diabetes are not sustainable since results from the current study shows similar effect in wound healing in DM and non-DM mice. The only reasonable conclusion is that microRNA-221-3p affects wound healing regardless the presence of DM or not.

Response: Thank you for your comment and question. Please allow us to combine these two questions to reply. First of all, we agree with your point that miRNA-221-3p can promote wound healing in both normal and diabetic mice. In previous results, diabetic mice did not have diabetes long enough to develop impaired wound healing, so we extended the diabetes duration (from the original 4 weeks to 12 weeks) in the new supplementary experiment in order to highlight the obvious promotion of miRNA-221-3p on wound healing in diabetic mice (text on lines 103-110, tracked changes). In addition, we represented the normal and diabetic wound healing on the same graph (**Fig.1a** for Representative photomicrographs and **Fig.1b, c** for summary data). We hope that the results of the new experiments can provide more information to demonstrate the promotion of miRNA-221-3p on wound healing in diabetes.

Comment 3: It would be beneficial to use also biomarkers for M1 and M2 macrophages, in addition to neutrophils and macrophages representative biomarkers that were used.

Response: Thank you very much for your constructive suggestion. We followed your advice and added the qPCR results of M1 and M2 macrophage to the revised manuscript. We labeled M1-type macrophages with *CD86*, and M2-type macrophages with *CD206*. The results are shown in **Fig. 2h** (text on lines 130-132, tracked changes).

Comment 4: It would be helpful to determine the protein levels of pro-inflammatory cytokines in addition to the mRNA expression.

Response: Thanks for your advice. Since inflammatory cytokines work primarily by secretion from cells, we chose to use enzyme-linked immunosorbent assay to test the collected supernatant from cells in order to

reflect the changes in the protein levels of inflammatory cytokines. The results are shown in **Fig. 3f-i** (text on lines 144-147, tracked changes).

Comment 5: More time points are needed for more comprehensive study of the expression levels of pro-inflammatory cytokines (not only day 11).

Response: We followed your suggestion to perform new experiments and added the results of the expression levels of inflammatory cytokines on day 1 and day 7 after wound formation in order to make study more comprehensive. The results are shown in **Fig. 2e-f** (text on lines 129-130, tracked changes).

Comment 6: A concentration 35mM Glucose used in the study is too high. A lower glucose concentration could provide more physiologically appropriate.

Response: We agree with your point that 35mM is too high level of glucose. Due to the complex pathological environment in vivo, maybe different patient has different glucose level and different localized tissue has different glucose level. Therefore, many studies have used different glucose to mimic different situation. For example, Rebecca Ward et al. used 50mM glucose to explore the relationship between hippocampal cells in diabetes and inflammasome^[1]. There are also relevant studies on keratinocytes and diabetic wound healing, Son DH et al. used 50mM glucose, found a novel peptide promotes antioxidant gene expression and wound healing in HaCaT Cells^[2]; Cheng TL et al. used 25mM and 50mM glucose found that high-glucose environment reduces the expression of Thrombomodulin in keratinocytes and released soluble Thrombomodulin can promote wound healing under diabetic condition^[3]. We tried to experiment with 25mM, but this glucose level was not enough to damage keratinocytes and cause increased expression of inflammatory factors. Finally, we chose 35mM glucose for further study and added equivalent amount of mannitol as a hyperosmolarity control to avoid the influence of high osmotic pressure on the experiment (**Fig. 3a**). We have added references to the revised manuscript to justify the glucose level.

Comment 7: For how many days did the miR-221-3p agomir and the miRNA mimic NC were injected to the wounds? It would be helpful if this information was added to the materials and methods section.

Response: Thanks for your suggestion, we have added the experimental schematic illustration for mice in the supplementary materials (**Supplementary Fig. 6.**) in order to make the experiment process more distinct. In addition, the specific dosage of the injected drug was shown on line 444-448 of the manuscript.

Comment 8: It would be beneficial to study/measure the re-epithelization and wound construction from MTS or H&E images, in addition the wound size measurements in fig. 1.

Response: According to your suggestion, we recollected samples and

repeated H&E staining in order to get higher quality photos to measure the re-epithelization and wound construction. Unfortunately, we did not obtain enough slices of the complete wound for statistical analysis, so the results from H&E staining were removed from the text. In addition, the wound size measurements were added into results (**Fig. 1c** and **Fig.7c**).

Comment 9: In lines 222-227 is mentioned that is described the Fig.5e—j. However, fig.5 does not have h, i and j, but stops at g. Please correct in the manuscript that in line 227 is described the fig. 5d-g.

Response: Thank you for pointing out this mistake, we have corrected it in the revised manuscript.

Comment 10: Author used DYRK1A-specific siRNA to suppress the expression of DYRK1A in vitro to further validate the impact of DYRK1A protein on inflammation and also showed the synergistic effect on inflammation reduction when apply both siRNA and miRNA. It looks like applying both siRNA and miRNA has a great potential, have author tried this combination or siRNA (solely) in vivo study?

Response: To verify this interesting finding, we conducted in vivo experiments in three groups: diabetic mice injected negative control, diabetic mice injected DYRK1A siRNA solely, and diabetic mice injected both DYRK1A siRNA and miR-221-3p agomir. Our results found that wound healing was significantly accelerated after application of DYRK1A siRNA solely (DM-si) and both application of siRNA and miR-221-3p agomir (DM-si+mi) compared with negative control (DM-Ctrl) (**Fig. 4j, k**). Unfortunately, we didn't get the results that applying both siRNA and miRNA create a more obvious effect. The experiment in vitro is relatively stable, but in vivo, physiological environment is more complex and many unpredictable factors can affect the experiment in vivo, So this superimposed effect failed observed in mice (text on lines 192-197, tracked changes).

Comment 11: Other studies have shown DYRK1A present in many different cell types, such as macrophage, fibroblasts, and endothelial cells, and have very different impact on angiogenic response. It would be helpful to add more explanation on why keratinocytes were selected in this study and what is the correlation between keratinocytes associated DYRK1A and DFU.

Response: We would like to explain the process in detail. First, our previous in vivo results showed that miRNA-221-3p significantly promoted wound healing in normal and diabetic mice. Wound healing is an extremely complex process in which a variety types of cells are involved. We further found in vivo that miRNA-221-3p can inhibit the inflammatory response of the skin at the edge of the wound. Therefore, keratinocyte (HaCaT), fibroblast (HSF), macrophage (RAW264.7/ THP-1) and endothelial cell (Huvec) were selected. We tested the expression of inflammatory cytokines in every types of cells after

overexpression of miRNA-221-3p, and finally found that the inflammatory response was significantly inhibited by miRNA-221-3p in keratinocytes. Keratinocyte is the major cellular component of epidermis and play critical roles in wound repair not only as structural cells but also exerting important immune functions. Second, after targeting the cell type, we intervene the cells with normal glucose or high glucose and transfected them with either negative control or miRNA-221-3p mimics, then the treated keratinocytes were sequenced by transcriptome. DYRK1A is one of seven candidate genes screened by bioinformatics prediction combined with transcriptome sequencing and finally verified in vivo and in vitro. In the last step, we conducted immunohistochemical staining with mouse skin, and the results showed that DYRK1A was mainly expressed in the epidermal layers of the skin, which was completely consistent with the previous cell experiment. To sum up: Starting from diabetic wound healing we found an inflammatory inhibitory effect of miRNA-221-3p, then we target keratinocytes and finally identified DYRK1A as the main target gene through sequencing and experimental verification.

Comment 12: Endothelial cells and fibroblasts are important for wound healing and affected by diabetes. It would be helpful to add more discussion on the effect on angiogenesis and collagen formation.

Response: Thank you very much for your constructive suggestion. We added discussion on the effects of other kinds of cells on wound healing, instead of just emphasizing keratinocytes (on lines 323-327, tracked changes). Further study will investigate whether miRNA-221-3p promotes wound healing through other pathways.

References

- [1] Ward R, Ergul A. Relationship of endothelin-1 and NLRP3 inflammasome activation in HT22 hippocampal cells in diabetes. *Life Sci.* 2016. 159: 97-103.
- [2] Son DH, Yang DJ, Sun JS, et al. A Novel Peptide, NicotinyI-Isoleucine⁺Valine⁻Histidine (NA⁺IVH), Promotes Antioxidant Gene Expression and Wound Healing in HaCaT Cells. *Mar Drugs.* 2018. 16(8).
- [3] Cheng TL, Lai CH, Chen PK, et al. Thrombomodulin promotes diabetic wound healing by regulating toll-like receptor 4 expression. *J Invest Dermatol.* 2015. 135(6): 1668-1675.

REVIEWERS' COMMENTS:

Reviewer #1 (Remarks to the Author):

the authors mostly addressed my questions and I have no more comments. The figures should still be polished as some of them have shifted legends.

Reviewer #4 (Remarks to the Author):

The authors have responded well to our questions and took all comments into consideration. It is appreciated that they extended the diabetes duration from 4 weeks to 12 weeks and re-conducted the experiments as suggested. They also added the biomarkers for 1 and 2 macrophage phenotype and included two more time points to observe the expression levels of pro-inflammatory cytokines in different stage of wound healing. Their explanation to why keratinocytes were specifically selected is valid. Finally, the additional discussion on other cells types associated with wound healing, instead of just emphasizing keratinocytes, is good; however, it would be more comprehensive if the fibroblasts were discussed as previously suggested.

There are no more questions or comments

Point-to-point Response

General response: We kindly thank reviewer for your constructive suggestions. In response to your comments, we have followed your suggestions to address these concerns. We believe that this version of the manuscript is largely improved.

Comment: The authors mostly addressed my questions and I have no more comments. The figures should still be polished as some of them have shifted legends.

Response: Thank you for pointing out this critical issue. We have revised the figures and figure legends to make the manuscript more accurate. All traces of revisions are marked in red and remain in the revised manuscript.

Point-to-point Response

General response: We kindly thank reviewer for your constructive suggestions. In response to your comments, we have followed your suggestions to revise the manuscript carefully. We believe that this version of the manuscript is largely improved.

Comment 1: The authors have responded well to our questions and took all comments into consideration. It is appreciated that they extended the diabetes duration from 4 weeks to 12 weeks and re-conducted the experiments as suggested. They also added the biomarkers for 1 and 2 macrophage phenotype and included two more time points to observe the expression levels of pro-inflammatory cytokines in different stage of wound healing. Their explanation to why keratinocytes were specifically selected is valid. Finally, the additional discussion on other cells types associated with wound healing, instead of just emphasizing keratinocytes, is good; however, it would be more comprehensive if the fibroblasts were discussed as previously suggested.

Response: Thank you very much for your constructive suggestion. We added discussion on the effects of fibroblasts on wound healing in the hope that this revision will make the discussion more comprehensive (on lines 313-314, tracked changes).